# Neuropeptide ACP facilitates lipid oxidation and utilization during long-term flight in locusts

Li Hou[1], Siyuan Guo[1], Yuanyuan Wang[1,2], Xin Nie[1], Pengcheng Yang[3], Ding Ding[1], Beibei Li[1], Le Kang[1]*, Xianhui Wang[1]*

[1]State Key Laboratory of Integrated Management of Pest Insects and Rodents, Institute of Zoology, Chinese Academy of Sciences, Beijing, China; [2]CAS Center for Excellence in Biotic Interactions, University of Chinese Academy of Sciences, Beijing, China; [3]Beijing Institutes of Life Science, Chinese Academy of Sciences, Beijing, China

**Abstract** Long-term flight depends heavily on intensive energy metabolism in animals; however, the neuroendocrine mechanisms underlying efficient substrate utilization remain elusive. Here, we report that the adipokinetic hormone/corazonin-related peptide (ACP) can facilitate muscle lipid utilization in a famous long-term migratory flighting species, *Locusta migratoria*. By peptidomic analysis and RNAi screening, we identified brain-derived ACP as a key flight-related neuropeptide. *ACP* gene expression increased notably upon sustained flight. CRISPR/Cas9-mediated knockout of *ACP* gene and ACP receptor gene (*ACPR*) significantly abated prolonged flight of locusts. Transcriptomic and metabolomic analyses further revealed that genes and metabolites involved in fatty acid transport and oxidation were notably downregulated in the flight muscle of ACP mutants. Finally, we demonstrated that a fatty-acid-binding protein (FABP) mediated the effects of ACP in regulating muscle lipid metabolism during long-term flight in locusts. Our results elucidated a previously undescribed neuroendocrine mechanism underlying efficient energy utilization associated with long-term flight.

*For correspondence:
lkang@ioz.ac.cn (LK);
wangxh@ioz.ac.cn (XW)

Competing interests: The authors declare that no competing interests exist.

## Introduction

Flight is an extraordinary biological trait that has only evolved in several kinds of animals (e.g. insects, bats, and birds) and is notably effective in searching for food and mates, finding habitats, defending against predators, and adapting to seasonal changes in environment (*Krause and Godin, 1996*; *Borgemeister et al., 1997*; *Chapman et al., 2010*). Despite these adaptive advantages, flight is one of the most intense and energy-demanding physiological processes, especially for insects that possess long-term flight capacity, such as locusts and butterflies (*Mentel et al., 2003*; *Zhan et al., 2014*). Long-term flight is usually defined as sustained flight for seasonal and long-range migration toward a distinct direction in populations (*Stefanescu et al., 2013*; *Juhász et al., 2021*). Metabolic rates in the flight muscle can increase by 20- to 100-fold during flight (*Harrison and Roberts, 2000*; *Suarez, 2000*). To meet the high energy demands of long-term flight, migratory insects have evolved a suite of adaptive physiological traits (*Arrese and Soulages, 2010*) and exhibit highly efficient utilization of energy substrates, such as carbohydrates and lipids, in the flight muscle (*Canavoso et al., 2003*; *Van der Horst and Rodenburg, 2010*). Compared to short-term flight, which exclusively employs carbohydrates as energy substrates, many physiological activities subsequently occur in the flight muscle during long-term flight, including carbohydrate/lipid metabolism transition, fatty acid transport, lipid oxidation, and lipid mobilization in the fat body. Clearly, complex and precise spatial and temporal regulation of energy metabolism is essential for long-term

**eLife digest** Flight allows insects to find food or seek a better environment. Some insects have developed the ability of 'long-term flight', which allows them to make continuous journeys over large distances. For example, one locust species regularly crosses the Red Sea which is up to 300 km wide – a spectacular feat for insects only a few inches long.

However, flight is an energy-intensive activity, and insects' muscles need the right sort of chemical fuel to work properly. Previous work has shown that this 'fuel consumption' is highly dynamic and happens in two stages. First, immediately after take-off, the muscles rapidly consume carbohydrates (sugars); then, during the prolonged phase of the flight, muscles switch to exclusively consume lipids (fats).

How the flight muscles 'know' when to start using fats for energy remains largely unclear. It has been suggested that this switch may involve hormone-like chemicals made in the brain called neuroendocrine peptides. Hou et al. therefore set out to test this hypothesis, using the locust species *Locusta migratoria* as a representative migratory insect.

Initial experiments used an abundance detection technique to determine which of the neuroendocrine peptides were active in adult locusts. Further analysis, looking specifically at locusts that had just been flying, revealed that the gene for a peptide called ACP became much more active after one hour of continuous flight. Further evidence that the ACP hormone could indeed be helping to power long-term flight came from locusts with a mutated, 'switched-off' version of the gene. These insects could only fly for half the time, and half the distance, compared to locusts that did not have mutations in the gene for ACP.

Biochemical studies of the ACP mutant locusts confirmed that their flight muscle cells could not transport and break down fatty acids normally. These experiments also showed that ACP was acting through a type of carrier protein called FABP, which is present in many different insects and normally 'ferries' lipids to the places they are needed.

These findings shed new light on the biological mechanisms that control long-term flight in migratory insects. The ability to move over long distances is key to the outbreak of locust plagues, which in turn cause widespread crop damage around the world. Hou et al. therefore hope that this knowledge will one day help develop effective strategies for locust pest control.

fight performance. However, the mechanisms underlying the highly efficient energy utilization associated with long-term flight have not been elucidated to date.

A number of reports have emphasized the central roles played by neuropeptides in coordinating systematic energy metabolism during insect flight (*Gade, 1992*; *Lorenz and Gäde, 2009*). Most neuropeptides are produced by the central nervous system and perform distinct tasks through binding with their cognate G-protein-coupled receptors (GPCRs) (*Nässel, 2002*). By affecting energy metabolism either directly or indirectly, neuropeptides participate in a variety of biological events, such as flight, reproduction, diapause, and immune response (*Gäde and Marco, 2009*; *Sim and Denlinger, 2013*; *Ling et al., 2017*; *Urbanski and Rosinski, 2018*). Furthermore, a single behavior or physiological process is usually controlled by multiple neuropeptides that play distinct roles in energy metabolism (*Waterson and Horvath, 2015*; *Toprak, 2020*). For example, adipokinetic hormone (AKH) has been demonstrated to be a conserved regulator of flight-related energy metabolism by promoting glycolysis and lipid mobilization in the fat body in different insect species (*Gäde et al., 2006*; *Kaufmann and Brown, 2008*). Downstream signal transduction of AKH involved in lipid mobilization has been elucidated in insects (*Gäde and Auerswald, 2003*). In addition, other neuropeptides are also involved in either lipogenesis or lipolysis and play distinct roles in insect flight (*Toprak, 2020*).

The migratory locust, *Locusta migratoria*, which possesses a notable long-term flight capacity, is one of the most destructive agricultural pests (*Wang and Kang, 2014*) and has been employed as a useful study model for the neurohormonal regulation of flight-related energy metabolism (*Jutsum and Goldsworthy, 1976*; *Van der Horst and Rodenburg, 2010*; *Bullard et al., 2017*). The locust displays strong adaption to long-distance flight at both the physiological and morphological levels, exhibiting clear expansion of the energy gene family and high plasticity of muscle metabolism (*Wang et al., 2014*). In the locust, the patterns of energy metabolism display tissue-specific and

time-dependent patterns during tethered flight. Usually, metabolism involved in long-term flight in insects contains two major phases: during the early stage of flight, carbohydrates in flight muscle and hemolymph significantly decrease, and lipid mobilization in the fat body gradually increases thereafter followed by utilization in the flight muscle during the prolonged flight phase (*Worm and Beenakkers, 1980*; *Wegener et al., 1986*), indicating a clear transition in energy consumption from carbohydrates to lipids during long-term flight. Distinct neuropeptide and neurotransmitter have been demonstrated to modulate lipid mobilization and transport in the fat body (*Van der Horst, 2003*) and carbohydrate catabolism in flight muscle at the beginning of flight (*Mentel et al., 2003*), and the relevant regulatory mechanisms have also been uncovered. Nevertheless, neuroendocrine mechanisms underlying energy utilization associated with prolonged flight remain to be explored.

In this study, we employed integrated multi-omics studies to screen potential neuropeptides involved in the long-term flight of locusts and elucidated relevant regulatory mechanisms by using the CRISPR/Cas9 method. Finally, we identified a novel flight modulator, the AKH/corazonin-related peptide (ACP), which plays an important role in prolonged flight by facilitating lipid utilization in the flight muscle in locusts.

## Results

### Identification of candidate neuropeptide regulators involved in the flight activity of locusts

Given that flight is a unique biological trait in adult locusts, we first performed comparative neuro-peptidome analysis on the neuroendocrinal tissues brain (Br) and retrocerebral complex (main endocrine tissues of the locust, RC) between final instar nymphs (5th) and adult locusts through high-resolution and high-sensitivity MS (LTQ-Orbitrap Elite) to identify neuropeptides possibly related to flight. In total, 201 and 362 nonredundant peptides (including both mature neuropeptides and their potential degradation products) derived from 37 neuropeptide precursors were identified in Br and RC, respectively (*Supplementary file 1*). Tissue-specific analysis showed that neuropeptides from 20 precursors were considerably more abundant in the RC, whereas neuropeptides from 16 precursors were more abundant in the Br. The GPB5-derived peptides showed similar abundance levels in Br and RC (*Figure 1—figure supplement 1*).

The abundant levels of neuropeptides in Br and RC between 5th-instar nymphs and adult locusts were further compared by a label-free quantitative strategy. Compared to 5th-instar nymphs, there were 20 and 18 upregulated neuropeptides in the Br and RC of adult locusts, respectively (*Figure 1—figure supplement 2*), and 10 neuropeptides displayed significantly higher abundance ($Log_2FC > 1.5$) in either Br or RC of adult locusts (*Figure 1A and B*). To validate whether these neuropeptides were closely related to flight activity, we examined the expression levels of the precursor genes of these ten neuropeptides in either Br or RC (depending on their tissue-specific expression patterns, *Figure 1—figure supplement 3*) after 1 h-sustained flight. The mRNA levels of four neuropeptide precursor genes, namely, AKH/corazonin-related peptide (*ACP*), adipokinetic hormone (*AKH2*), *NPF1*, and *GPB5*, changed significantly after flying treatment. Among these genes, three (*ACP*, *AKH2*, and *NPF1*) exhibited clearly increased expression levels, whereas *GPB5* displayed decreased expression levels after 1 h- of sustained flight (*Figure 1C* and *Figure 1—figure supplement 4*).

To explore whether these four neuropeptides are involved in the modulation of flight performance in locusts, we performed gene knockdown by RNA interference (RNAi) for each gene in adult locusts. Of these four genes, only the *ACP* and *AKH2* RNAi treatments exhibited significant effects on the flight activity of adult locusts, although the expression levels of all neuropeptide genes were successfully downregulated (*Figure 1—figure supplement 5*). Compared with the control, total flight time and total flight distance decreased by more than 50% after the knockdown of *ACP* or *AKH2* (*Figure 1D and E*). However, the average flight velocity and maximum flight velocity of the locusts did not significantly change (*Figure 1—figure supplement 6A and B*). AKH family members have been determined to play conserved roles in flight activity by promoting the mobilization of lipids and carbohydrates stored in the fat body of locusts (*Van der Horst, 2003*). However, to date, few studies have investigated ACP functions in insects. Thus, we then validated the regulatory role of ACP in flight activity by injecting synthetic ACP peptide in adult locusts. After peptide injection,

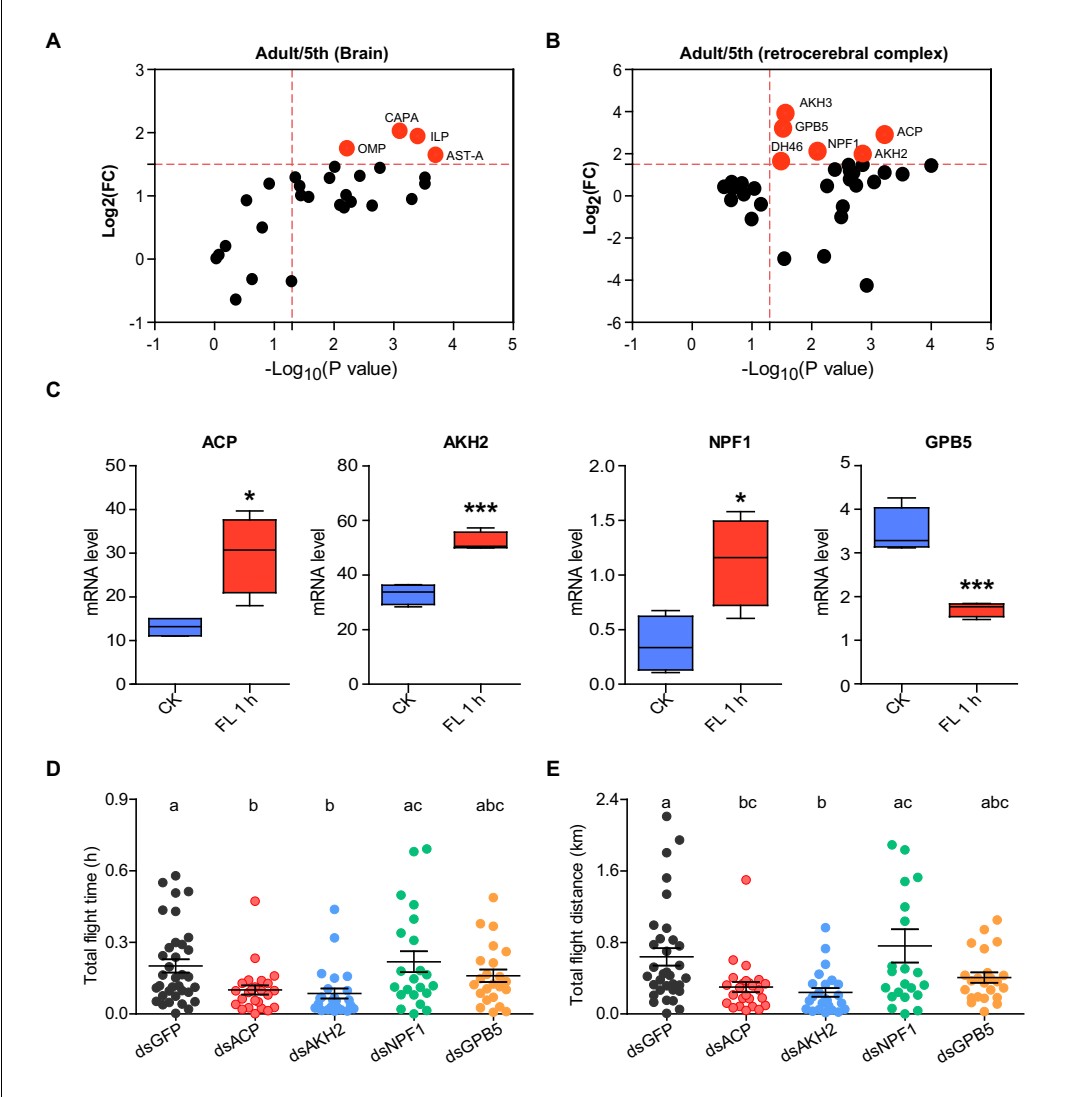

**Figure 1.** Identification of AKH/Corazonin-related neuropeptide (ACP) as a potential neuroendocrine modulator for flight activity in the locust. Volcano plot of neuropeptidomes from (**A**) brains (Br) and (**B**) the retrocerebral complex (RC) of the final instar nymphs (5th) and mature adult locusts. Each circle represents a neuropeptide. Differential peptides with a $Log_2$(FC) >1.5 and p value < 0.05 are highlighted in red. Data are from three biological replicates. (**C**) Expression levels of four neuropeptide precursor genes (*ACP*, *AKH2*, *NPF1*, and *GPB5*) in Br or RC after 1 h-sustained tethered flight (Student's t-test, p=0.011, t = 3.619, df = 6 for *ACP*; *p*=0.0003, t = 7.394, df = 6 for *AKH2*; *p*=0.0194, t = 3.166, df = 6 for *NPF1*; *p*=0.0007, t = 6.451, df = 6 for *GPB5*, respectively, n = 4 biological replicates for each treatment, *p<0.05, ***p<0.001). CK indicates control, FL 1 h indicates 1h-sustained flight. (**D**) Total flight time and (**E**) total flight distance after the knockdown of four candidate neuropeptide precursor genes, *ACP*, *AKH2*, *NPF1*, and *GPB5*, in adult locusts. Columns labeled with different letters indicate that there is a significant difference between the two groups, columns contain same letters indicate no significance observed between the two groups (one-way ANOVA for D, *F* = 4.658, df = 4, p=0.0016; for E, *F* = 5.48, df = 4, p=0.0004; n = 30 (dsGFP), 25 (dsACP), 24 (dsAKH2), 22 (dsNPF1), and 23 (dsGPB5), respectively). Each blot represents a single individual. The mean ± SEM are shown. See *Figure 1—source data 1* for details.

The online version of this article includes the following source data and figure supplement(s) for figure 1:

**Source data 1.** Raw data for comparisons of peptide contents, gene expression, and flight activity.
**Figure supplement 1.** Tissue-specific abundance of neuropeptides identified in the neuropeptidome.
**Figure supplement 2.** Comparison of neuropeptide abundance in Brain and retrocerebral complex between 5th instar nymphs and adult locusts.
**Figure supplement 3.** Expression levels of 10 neuropeptide precursor genes in Brain and retrocerebral complex of adult locusts.
**Figure supplement 4.** Expression levels of six neuropeptide precursor genes in Brain or retrocerebral complex upon 1 h-sustained flighting.
**Figure supplement 5.** RNAi efficiency of four flight-related neuropeptide precursor genes examined by qPCR.
**Figure supplement 6.** Measurement of (A) average flight velocity and (B) maximum flight velocity after the knockdown of four candidate neuropeptide precursor genes, *ACP*, *AKH2*, *NPF1*, and *GPB*, in adult locusts.
*Figure 1 continued on next page*

*Figure 1 continued*

**Figure supplement 7.** Measurement of (A) total flight time, (B) total flight distance, (C) average flight velocity, and (D) maximum flight velocity after the injection of synthetic ACP peptide in wide type locusts.

the locusts exhibited significantly enhanced total flight time and total flight distance, whereas average and maximum flight velocity were unaffected (*Figure 1—figure supplement 7*). These results confirm the essential regulatory role of ACP peptide in locust flight ability.

## Mutant line of CRISPR/Cas9 confirmed the essential role of *ACP* in long-term flight

The *ACP* precursor gene exhibited a brain-specific expression pattern (*Figure 2A*). In addition, the mRNA level of *ACP* did not significantly change until sustained flight for 1 h, with a 100% increase being observed at this time point (*Figure 2B*), implying that the ACP peptide plays major roles in facilitating long-term flight in locusts. To further explore the functional roles played by ACP in long-term flight, we generated an *ACP* mutant line using a CRISPR/Cas9-mediated gene editing system. We injected Cas9 protein and a gRNA targeting the second exon of the *ACP* gene into eggs < 2 h after they were laid (*Figure 2C*). The *ACP* gRNA-induced mutation at high efficiency in the G0 generation, as well as their progeny (G1) (*Figure 2—figure supplement 1A*, *Supplementary file 2*). We finally successfully obtained a heritable homozygous *ACP* mutant line with a 13 bp deletion modification (ACP[13/13], referred to as ACP[-/-] in the following text) by performing a series of crossing experiments (*Figure 2D* and *Figure 2—figure supplement 1B*). The ACP[-/-] locusts were predicted to produce a frameshift precursor unable to give rise to mature ACP peptide. Through immunohistochemistry analysis, we detected strong signals for ACP peptide in the neurons of the par intercerebralis and bilateral forebrain of wild-type locusts (WT) (*Figure 2E* and *Figure 2—figure supplement 2*), whereas the fluorescence signal was not observed in the brain of ACP[-/-] locusts (*Figure 2E*), which further confirmed the successful construction of *ACP* mutants of the migratory locust.

In comparison with WT locusts, no significant difference in the survival rate of either females or males was observed (*Figure 2—figure supplement 3*), whereas ACP[-/-] locusts had a larger body size (*Figure 2F*). Next, we compared the flight performance of ACP mutants and WT locusts during the 60 min sustained tethered flight test. Within the first 15 min, there was no significant difference in total flight distance and total flight time between female ACP mutants and WT locusts. However, for the 60 min tethered flight test, both females and males of ACP mutants displayed significantly shorter flight times and flight distances compared with the WT locusts (*Figure 2G and H*). The average flight velocity and maximum flight velocity did not show any changes between ACP mutants and WT locusts in either 15 min or 60 min flight tests (*Figure 2I* and *Figure 2—figure supplement 4*). These results indicated that the neuropeptide ACP was involved in modulating long-term flight in locusts.

## Loss of function of ACP receptor (ACPR) significantly impaired long-term flight in the locust

Distinct receptors play essential roles in mediating the functions of neuropeptides, and tissue-specific expression patterns of neuropeptide receptors have been proposed to be sources of critical information for exploring the regulatory mechanisms of neuropeptides (*Garcia et al., 2015*; *Nässel and Vanden Broeck, 2016*). Therefore, we further identified the ACP receptor by homolog searching the genome and transcriptome databases. Phylogenetic analysis showed that the putative locust ACPR was closely related to ACP receptors from other insects but was evolutionarily divergent from its structure-related neuropeptide receptors, AKH receptors and corazonin receptors (*Figure 3A*). Compared to other organs tested, *ACPR* was highly expressed in the fat body and flight muscle of adult locusts (*Figure 3B*). To validate the role of ACPR in regulating long-term flight, we generated an *ACPR* mutant locust line using the CRISPR/Cas9 system. Combined injection of designed *ACPR* gRNA and Cas nine protein induced multiple kinds of mutations around the target sequence in the G0 generation (*Figure 3C* and *Supplementary file 3*). Using the crossing strategy similar to *ACP* mutant line construction, we successfully obtained a homozygous *ACPR* mutant line with a 13 bp deletion (ACPR[13/13]). The ACPR[13/13] locusts were predicted to produce truncated

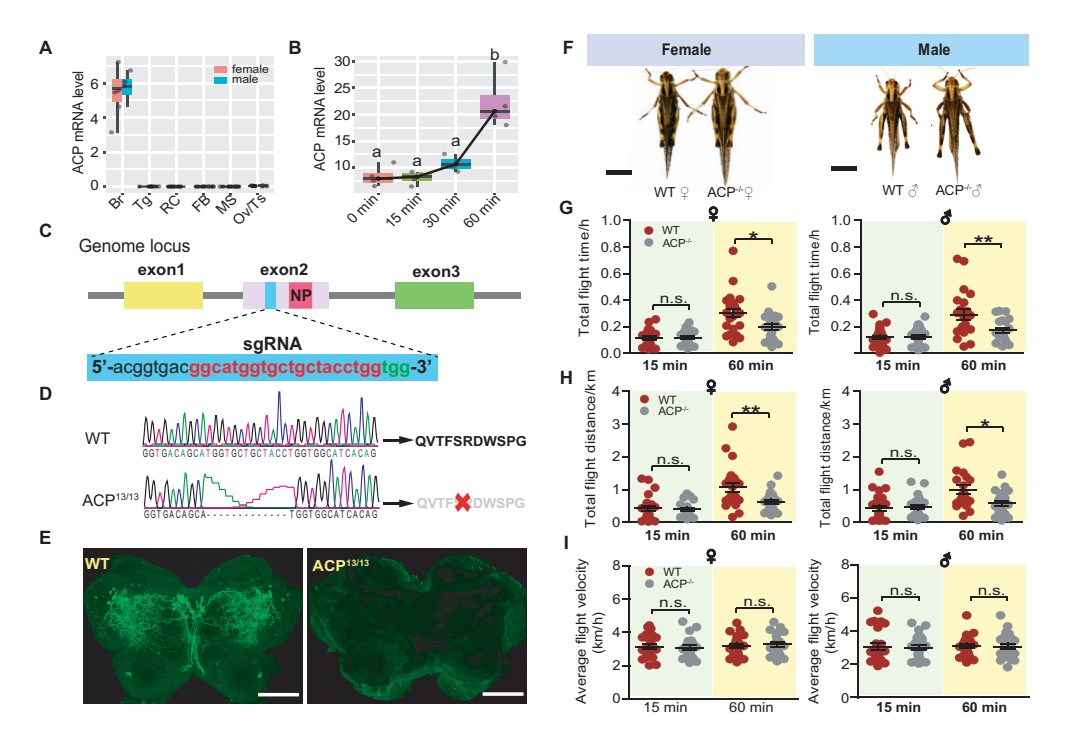

**Figure 2.** Functional role of ACP in long-term flight validated by CRISPR/Cas9 system-mediated gene knockout. (**A**) Tissue-specific expression pattern of the *ACP* precursor gene in adult locusts analyzed by qPCR (n = 4 replicates, 6–8 locusts/replicate). Br, brain; RC, retrocerebral complex; FB, fat body; MS, muscle; Ov, ovary; Tg, thoracic ganglia. (**B**) Expression levels of the *ACP* gene in the brain during the time course of sustained flighting of adult locusts. The mRNA level of *ACP* significantly increased after 1 h of sustained flighting. Different letters indicate that there is a significant difference between the two groups (one-way ANOVA, F = 20.69, df = 3, p<0.0001, n = 4 replicates). (**C**) Gene structure and designed sgRNA-targeted site in exon 2 of the locust *ACP* gene. Three exons are represented in different colors. The sgRNA targeted site in exon two is highlighted in blue. The pink region in exon two indicates the mature ACP peptide-encoding region. (**D**) Genome PCR product sequencing of the wild-type (WT) and 13 bp-deletion ACP mutants (ACP[13/13], referring to ACP[-/-] in the following section), which could not give rise to mature ACP peptide. (**E**) Detection of ACP peptide in WT and ACP[-/-] locusts by immunohistochemistry assay. The bar indicates 100 µm. (**F**) Morphology of females and males of WT and ACP[-/-] locusts. Both females and males of ACP mutant adults show increased body size compared to that of WT controls. Bars represent 1 cm. Measurement of (**G**) total flight time, (**H**) total flight distance, and (**I**) average flight velocity in females and males of WT and ACP[-/-] locusts. Each dot represents a single individual (Student's t-test for G, p=0.8499, t = 0.1904, df = 44 for 15 min♀, p=0.0183, t = 2.460, df = 40 for 60 min♀; p=0.9504, t = 0.0626, df = 46 for 15 min♂, p=0.0088, t = 2.755, df = 40 for 60 min♂; For H, p=0.6416, t = 0.4687, df = 44 for 15 min♀, p=0.0097, t = 2.718, df = 40 or 60 min♀; p=0.7489, t = 0.332, df = 46 for 15 min♂, p=0.01, t = 2.704, df = 40 for 60 min♂; For I, p=0.6067, t = 0.5186, df = 44 for 15 min♀, p=0.654, t = 0.4516, df = 40 for 60 min♀; p=0.7644, t = 0.3015, df = 46 for 15 min♂, p=0.9122, t = 0.1109, df = 40 for 60 min♂; n = 23 (WT♀ 15 min), 23 (ACP[-/-]♀ 15 min), 22 (WT♀ 60 min), 20 (ACP[-/-]♀ 60 min), 24 (WT♂ 15 min), 24 (ACP[-/-]♂ 15 min), 21 (WT♂ 60 min), 21 (ACP[-/-]♂ 60 min), *p<0.05, **p<0.01, n.s. indicates not significant). See *Figure 2—source data 1* for details.

The online version of this article includes the following source data and figure supplement(s) for figure 2:

**Source data 1.** Raw data for gene expression and flight activity.

**Figure supplement 1.** Construction of the ACP loss-of-function mutant line of the migratory locust.

**Figure supplement 2.** Localization of ACP peptide in the locust brain detected by IHC.

**Figure supplement 3.** Survival rate of females and males of WT and ACP[-/-] locusts.

**Figure supplement 4.** Measurement of maximum flight velocity in (**A**) females and (**B**) males of WT and ACP[-/-] locusts.

protein that losses the last four transmembrane domains (*Figure 3D*). Compared with the WT locusts, both ACPR female and male mutants showed significantly reduced flight time and flight distance during the 60 min tethered flight test (*Figure 3E and F*), although intense immunostaining signal of ACP peptide was detected in the brain of ACPR mutants (*Figure 3—figure supplement 1*). However, no significant changes in average flight velocity and maximum flight velocity were observed between ACPR mutants and WT locusts (*Figure 3—figure supplement 2*). The flight phenotypes caused by *ACPR* knockout was similar to that observed in ACP mutants, supporting that the essential role of ACP peptide system in modulating long-term flight in locusts.

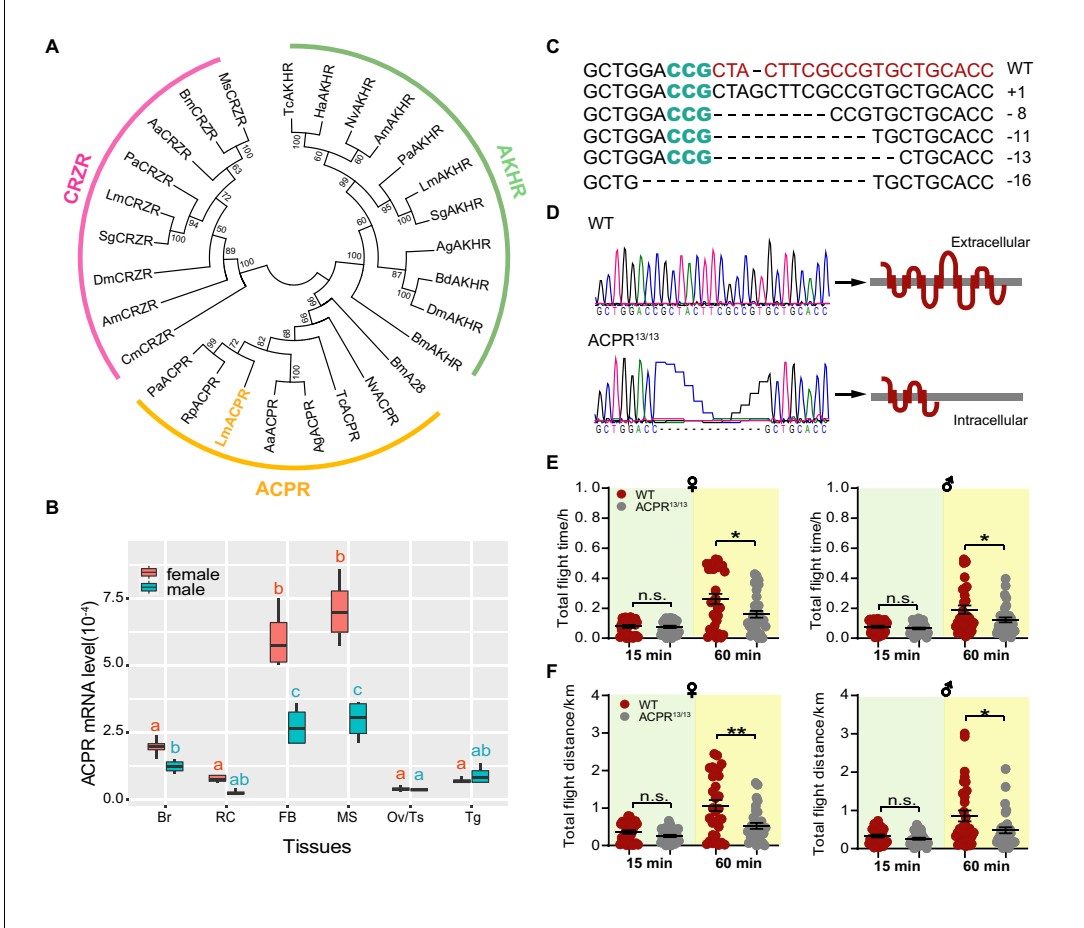

**Figure 3.** CRISPR/Cas9-mediated knockout of *ACPR* impairs long-term flight in the locust. (**A**) Phylogenic relationship of AKHR, ACPR, CRZR from the migratory locust and their homologs from other representative insect species. The phylogenetic tree is constructed using the neighbor-joining method. The locust ACPR protein is evolutionarily divided into the insect ACPR cluster. (**B**) Tissue-specific expression pattern of *ACPR* in both female and male adult locusts. The data are presented as the mean ± SEM (one-way ANOVA, *F* = 27.08, df = 5, p<0.0001 for male; *F* = 73.02, df = 5, p<0.0001 for female, n = 4 replicates, 6–8 locusts/replicate). For both female and male adults, *ACPR* gene is highly expressed in fat body and muscle, compared with other tissues tested. Columns labeled with different letters in same color indicate that there is a significant difference between the two groups, columns contain same letters in same color indicate no significance observed between the two groups (red indicates female, green indicates male). Br, brain; RC, retrocerebral complex; FB, fat body; MS, muscle; Ov, ovary; Tg, thoracic ganglia. (**C**) Gene types of the *ACPR* mutation in the G0 generation determined by Sanger sequencing. (**D**) Sequencing of the wild-type (WT) and 13 bp-deletion homozygous ACPR mutants (ACPR13/13), which is predicted to produce a truncated protein that lacks the last four transmembrane domains. Measurement of (**E**) total flight time and (**F**) total flight distance in females and males of WT and ACP-/- locusts. Each dot represents a single individual (Student's t-test for E, p=0.7536, t = 0.3153, df = 62 for 15 min♀, p=0.0157, t = 2485, df = 62 for 60 min♀; p=0.2481, t = 1.166, df = 63 for 15 min♂, p=0.0384, t = 2.115, df = 63 for 60 min♂; For F, p=0.0734, t = 1.821, df = 62 for 15 min♀, p=0.0016, t = 3.295, df = 62 or 60 min♀; p=0.1271, t = 1.546 df=63 for 15 min♂, p=0.0227, t = 2.336, df = 63 for 60 min♂; *p<0.05, **p<0.01, n.s. indicates not significant). See *Figure 3—source data 1* for details.

The online version of this article includes the following source data and figure supplement(s) for figure 3:

**Source data 1.** Raw data for gene expression and flight activity.
**Figure supplement 1.** Immunostaining of ACP peptide in the brain of ACPR13/13 locusts.
**Figure supplement 2.** Measurement of (**A**) average flight velocity and (**B**) maximum flight velocity in females and males of WT and ACPR13/13 locusts.

## Transcriptome analysis reveals significant downregulation of genes associated with lipid metabolism in the flight muscle of ACP$^{-/-}$ locust adults

Based on the tissue-specific expression pattern of *ACPR*, we hypothesized that fat body and flight muscle may be the main tissues targeted by ACP to participate in flight regulation. We then performed comparative transcriptome analysis of the fat body and flight muscle tissues in ACP$^{-/-}$ and WT locusts. The results showed that the number of differentially expressed genes (DEGs) in flight muscle was greater than that in fat body (520 in flight muscle and 318 in fat body, Log$_2$ FC >1, FDR < 0.05, RPKM > 0.5). For ACP$^{-/-}$ locusts, there were 212 upregulated and 308 downregulated genes in the flight muscle and 200 upregulated and 118 downregulated genes in the fat body (*Figure 4—figure supplement 1*). The fat body and flight muscle had more tissue-specific DEGs and fewer overlapping DEGs after knockout of the *ACP* gene (*Figure 4A*). Several pathways associated with energy metabolism were significantly changed in the flight muscle of ACP$^{-/-}$ locusts (-Log$_2$(P value)>10, enriched gene number >15), including oxidation phosphorylation, fatty acid degradation, valine, leucine and isoleucine degradation, cardiac muscle contraction, and fatty acid metabolism. However, only a small number of genes were enriched in the KEGG analysis of the fat body (-Log$_2$(P value)<10, gene number <10) (*Figure 4B*), indicating that the gene expression profiles of the flight muscle were more strongly affected by *ACP* knockout.

The expression levels of 4 genes responsible for fatty acid transport, 10 genes involved in beta-oxidation, and 19 genes associated with mitochondrial energy metabolism were clearly downregulated in the flight muscle of the ACP$^{-/-}$ locust (*Figure 4C*). A fatty acid binding protein (FABP), as the most highly expressed gene (Log$_{10}$(mean expression)>4), exhibited an expression decrease of more than 80% (log$_2$FC = −2.43) in ACP$^{-/-}$ locusts (*Figure 4D*). Reduced expression levels of FABP in the flight muscle of ACP$^{-/-}$locusts were further confirmed by qPCR and western blot analyses (*Figure 4E and F*). We also validated decreased mRNA levels of eight other genes involved in fatty acid transport and beta-oxidation in the flight muscle of ACP mutants (*Figure 4—figure supplement 2A*). Reduced expression of representative genes that participate in mitochondrial fatty acid transport (CPT2) and beta-oxidation (ACDM) was also confirmed at the protein level (*Figure 4—figure supplement 2B*). Moreover, we found that the expression levels of genes related to fatty acid transport and oxidation were strongly enhanced in the flight muscle upon ACP peptide injection as well as 1 h-sustained flight of WT locusts (*Figure 4—figure supplement 3*), implying that the ACP peptide may facilitate long-term flight by promoting fatty acid utilization. Given that FABP serves as a primary transporter for fatty acid translocation through the aqueous cytosol to mitochondria, where the beta-oxidation process takes place (*Haunerland and Spener, 2004*), we further assessed whether the change in FABP could affect the expression levels of beta-oxidation-related genes. RNAi knockdown of *FABP* resulted in a 99.3% decrease in *FABP* mRNA levels (*Figure 4—figure supplement 4*) and clearly suppressed the expression levels of multiple beta-oxidation-related genes, including *CROT*, *CPT2*, *ACDM*, *ACADS*, *ACADSB*, *ECH-6*, *ACAT1*, and *CRAT* (*Figure 3G*). Taken together, these results indicated that the lipid metabolism pathway was significantly suppressed in the flight muscle of ACP$^{-/-}$ locusts and that FABP may serve as an important molecular target of ACP peptide signaling.

## Metabolome analysis indicates decreased lipid utilization in the flight muscle of ACP$^{-/-}$ locusts

Based on the above results, we hypothesized that *ACP* knockout may affect energy utilization in the flight muscle of locust adults. To verify this possibility, we performed comparative metabolome analysis between ACP$^{-/-}$ and WT flight muscles by ultra-high-performance liquid chromatography-high resolution mass spectrometry (UPLC-HRMS). In total, we identified 881 metabolites with confidence from all twelve samples (*Figure 5—figure supplement 1*). Locust samples of WT and ACP$^{-/-}$ can be clearly separated by unsupervised principal component analysis (PCA) and orthogonal projection to latent structures–discriminant analysis (OPLS-DA) (*Figure 5A* and *Figure 5—figure supplement 2*). The overall metabolite distribution was considerably more intense in WT samples than in ACP$^{-/-}$ samples (*Figure 5A*), suggesting reduced general metabolic activity in the flight muscle of ACP$^{-/-}$ locusts.

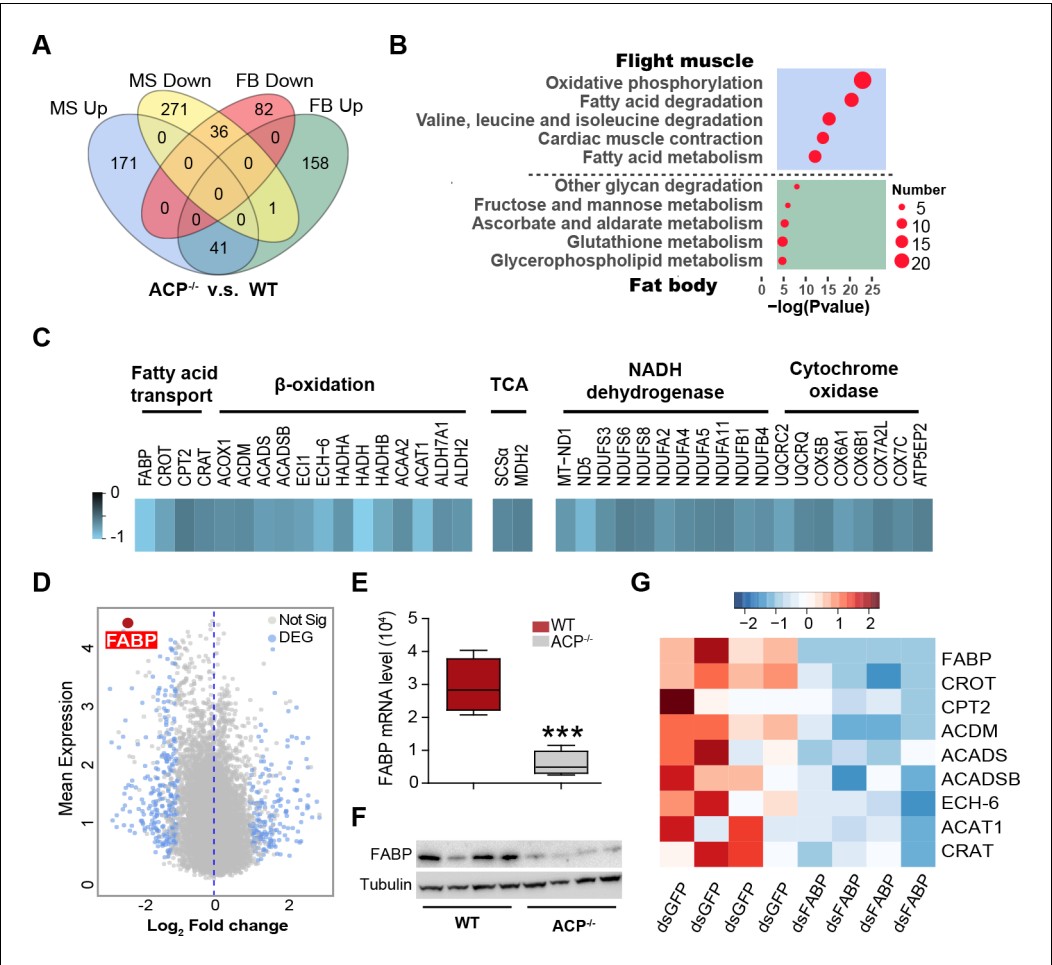

**Figure 4.** Transcriptome analysis reveals significant downregulation of genes and pathways associated with lipid transport and oxidation in the flight muscle of ACP[-/-] locusts. (**A**) Ven diagram of differentially expressed genes (DEGs) in flight muscle and fat body between WT and ACP[-/-] locusts. There were 212 upregulated and 349 downregulated genes in the flight muscle and 200 upregulated and 118 downregulated genes in the fat body of ACP mutants. (**B**) The representative enriched Kyoto Encyclopedia of Genes and Genomes (KEGG) terms of the DEGs in the flight muscle and fat body in WT and ACP[-/-] locusts. (**C**) Heat map of DEGs related to fatty acid transport, beta-oxidation, and oxidation phosphorylation in the flight muscle of WT and ACP[-/-] locusts. (**D**) Volcano plot of RNA-seq data from flight muscle of WT and ACP[-/-] locusts. Blue dots indicate DEGs, and the red dot indicates fatty acid-binding protein (FABP) that shows highly basic expression and fold change after *ACP* gene knockout. Validation of FBAP expression in the flight muscle of WT and ACP[-/-] locusts via (**E**) qPCR and (**F**) western blot. (Student's t-test for E, p=0.0007, t = 5.756, df = 7, n = 4–5 biological replicates, ***p<0.001). (**G**) Heat map of beta-oxidation-related genes in flight muscle after *FABP* knockdown. See *Figure 4—source data 1* for details. The online version of this article includes the following source data and figure supplement(s) for figure 4:

**Source data 1.** Raw data for RNA-seq and gene expression.
**Figure supplement 1.** Numbers of differentially expressed genes in the flight muscle and fat body between WT control and ACP[-/-] locusts.
**Figure supplement 2.** Expression levels of genes related to fatty acid transport and beta-oxidation are downregulated in the flight muscle of ACP[-/-] locusts.
**Figure supplement 3.** Expression levels of genes related to fatty acid transport and oxidation after ACP peptide injection and 1 h-sustained flight in WT locusts.
**Figure supplement 4.** RNAi efficiency of FABP in the muscle examined by qPCR.

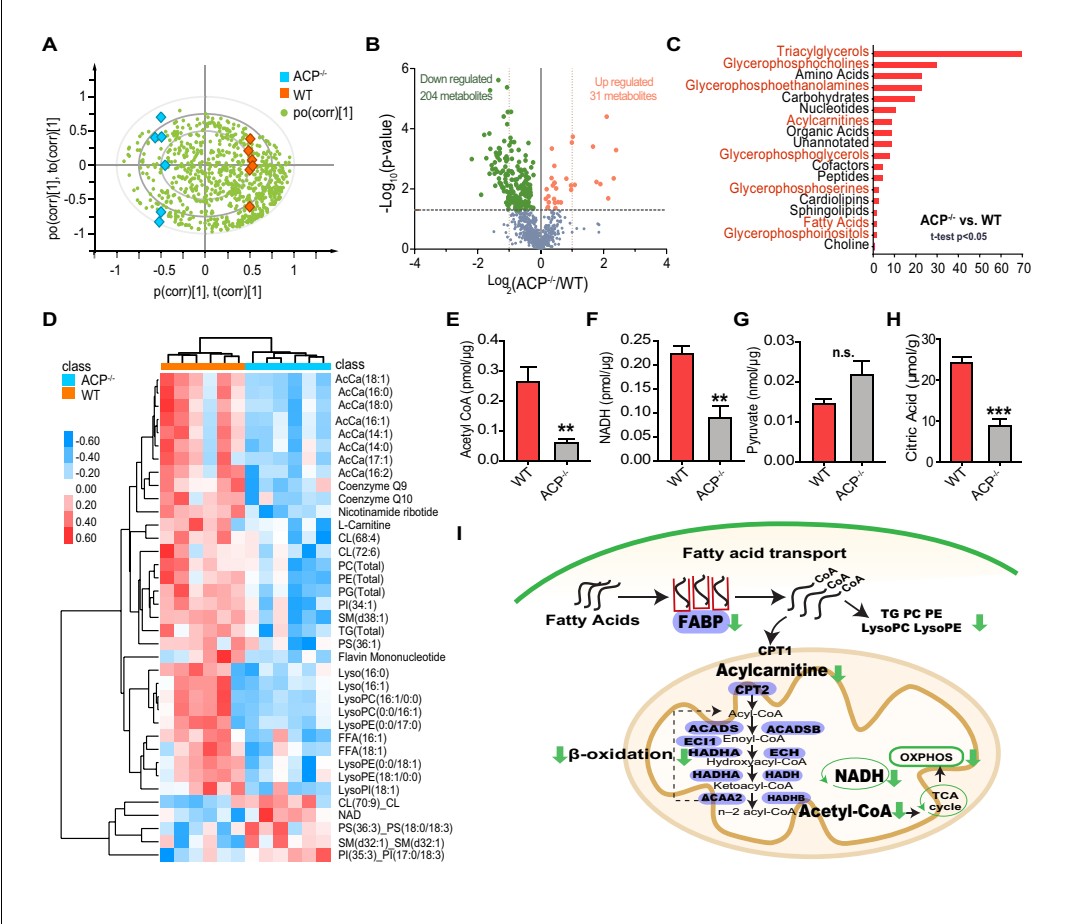

**Figure 5.** Metabolomic analysis reveals impaired lipid metabolism in the flight muscle of ACP mutants. (**A**) Metabolite distribution in the flight muscle of WT and ACP$^{-/-}$ locusts, as determined by orthogonal partial least squares discriminant analysis (OPLS-DA). Red blocks indicate WT samples, blue blocks indicate ACP$^{-/-}$ samples. Each green dot indicates a single metabolite. (**B**) Volcano plot of the metabolome from flight muscle of WT and ACP$^{-/-}$ locusts. There were 31 upregulated and 204 downregulated metabolites in the flight muscle of ACP mutants compared to WT locusts. Metabolites with p values < 0.05 are highlighted in red (upregulated) and green (downregulated). (**C**) Chemical structure classification of differential metabolites between WT and ACP$^{-/-}$ samples. (**D**) Heat map of differential metabolites related to lipid metabolism, including acylcarnitine, triglyceride, phospholipid, and coenzyme. The contents of metabolites are highlighted in red (upregulated) and blue (downregulated). Abundance detection of (**E**) acetyl CoA, (**F**) NADH, (**G**) pyruvate, and (**H**) citric acid in the flight muscle of WT and ACP$^{-/-}$ locusts. (Student's t-test for E, p=0.007, t = 3.472, df = 9, n = 6 (WT) and 5 (ACP$^{-/-}$); for F, p=0.0022, t = 4.439, df = 8, n = 5 (WT) and 5 (ACP$^{-/-}$); for G, p=0.0857, t = 1.959, df = 8, n = 5 (WT) and 5 (ACP$^{-/-}$); for H, p=0.0001, t = 7.085 df=8, n = 5 (WT) and 5 (ACP$^{-/-}$), **p<0.01, ***p<0.001). (**I**) Schematic diagram of the metabolic pathway combining the metabolomic and transcriptome analyses. The metabolites involved in fatty acid transport and subsequent oxidation are downregulated in the flight muscle of ACP mutants compared to WT locusts. Genes associated with lipid transport and oxidation are also presented (highlighted in purple). See **Figure 5—source data 1** for details.

The online version of this article includes the following source data and figure supplement(s) for figure 5:

**Source data 1.** Raw data for metabolic and metabolite contents.

**Figure supplement 1.** Pie chart of metabolite chemical classes structurally annotated in this experiment.

**Figure supplement 2.** Unsupervised PCA score plots of metabolic phenotypes between the ACP$^{-/-}$ and WT groups.

**Figure supplement 3.** Heat map of TG, PC, and PE in the flight muscle of WT and ACP$^{-/-}$ locusts.

**Figure supplement 4.** Measurements of acetyl CoA and NADH levels in the flight muscle after ACP peptide injection and 1 h-sustained flight in WT locusts.

Statistically, we obtained 204 downregulated metabolites and 31 upregulated metabolites in ACP$^{-/-}$ samples (Student's t-test, p<0.05) (**Figure 5B**). The differential metabolites identified between the two groups included many triacylglycerols (TG), glycerophosphatides (PC, PE, PG, LysoPC, LysoPE), amino acids, carbohydrates, acylcarnitine, and nucleotides (**Figure 5C** and **Figure 5—figure supplement 3**). Clustering analysis demonstrated that eight medium/long-chain acylcarnitines

(AcCa18:1, AcCa16:0, AcCa18:0, AcCa16:1, AcCa14:1, AcCa14:0, AcCa17:1, AcCa16:2) were present at significantly reduced levels in the flight muscle of ACP$^{-/-}$ locusts. Meanwhile, several coenzymes involved in mitochondrial oxidative phosphorylation were decreased, such as coenzyme Q9, coenzyme Q10, nicotinamide ribotide, and flavin mononucleotide (*Figure 5D*). These data indicated that acyl carnitine-dependent fatty acid transport to the mitochondrion and subsequent metabolism were notably decreased in the flight muscle of ACP mutants.

We further evaluated beta-oxidation status between WT and ACP$^{-/-}$ samples by determining the relative amounts of two end metabolites of beta-oxidation, acetyl-CoA and NADH. Relative amounts of acetyl-CoA and NADH in the ACP$^{-/-}$ samples significantly decreased by 77% and 65%, respectively, relative to the levels observed in WT locusts (*Figure 5E and F*). Instead, the injection of ACP peptide could significantly enhance the relative amounts of acetyl-CoA and NADH in the flight muscle of WT locusts (*Figure 5—figure supplement 4A*). However, upon 1 h-tethered fight, the relative content of acetyl-CoA did not change but the NADH level was decreased (*Figure 5—figure supplement 4B*). As a key intermediate metabolite of energy metabolism, acetyl-CoA is primarily generated from fatty acid oxidation and oxidative decarboxylation of pyruvate, which is an important product of glycolysis (*Rui, 2014*). We next assessed whether the glycolysis process was affected by *ACP* knockout by measuring the pyruvate amount. We observed that the relative level of pyruvate showed no significant difference between WT and ACP$^{-/-}$ samples (*Figure 5G*). To confirm the changes in energy metabolism in ACP$^{-/-}$ locusts, we also determined the abundance of citric acid, a representative metabolite of the tricarboxylic acid (TCA) cycle produced from acetyl-CoA (*Koubaa et al., 2013*). The relative level of citric acid in the WT samples was 2.82-fold higher than that in ACP$^{-/-}$ samples (*Figure 5H*). Therefore, by combined analysis of gene expressions and metabolite contents, we inferred that the downregulation of genes and metabolites involved in lipid transport and beta-oxidation in the flight muscle contributes to the deteriorated flight ability of ACP$^{-/-}$ mutant (*Figure 5I*).

## FABP mediates the regulatory effects of ACP on lipid metabolism and flight performance

We further verified whether FABP acts as a key downstream molecular target of ACP to regulate lipid metabolism in flight muscle during the long-term flight of locusts. First, we carried out a 60 min-tethered flight test after performing gene knockdown of *FABP* in locust adults. Compared to ds*GFP*-injected locusts, the total flight duration and total flight distance did not change within the first 15 min but were significantly decreased after 60 min of flight in ds*FABP*-injected locusts (*Figure 6A and B*). The average and maximus flight velocity did not change significantly, although a declining tendency was observed after *FABP* knockdown (*Figure 6—figure supplement 1*). Next, we measured the amount of acetyl-CoA and NADH because they indicate beta-oxidation status after ds*FABP* treatments. Compared to the ds*GFP* control, the relative levels of acetyl-CoA and NADH were strongly reduced by 70% and 44% in the ds*FABP*-injected samples (*Figure 6C and D*), respectively.

To demonstrate the key role played by FABP in mediating the effect of ACP on lipid metabolism in flight muscle, we performed a molecular rescue experiment by combined peptide injection and gene knockdown in ACP$^{-/-}$ locusts. When ACP peptide was injected in ACP$^{-/-}$ locusts, the expression levels of eight key beta-oxidation genes significantly increased in the flight muscle, whereas this stimulatory effect of ACP injection was remarkably abolished by *FABP* knockdown (*Figure 6E*). The knockdown of *FABP* also alleviated the upregulated amount of acetyl-CoA and NADH in the flight muscle (*Figure 6F and G*). In addition, the enhancement of the contents of multiple medium/long-chain acylcarnitines induced by ACP peptide administration also disappeared after *FABP* gene silencing (*Figure 6H*). In particular, the impaired prolonged flight performance in ACP$^{-/-}$ locusts, including reductions in both total flight duration and flight distance, could be efficiently recovered by ACP peptide injection. Moreover, this recovered flight activity induced by the ACP peptide was clearly blocked by ds*FABP* treatments (*Figure 6I*). Taken together, these results indicated that FABP acts as a key component of ACP signaling, regulating lipid metabolism of the flight muscle during long-term flight in locusts.

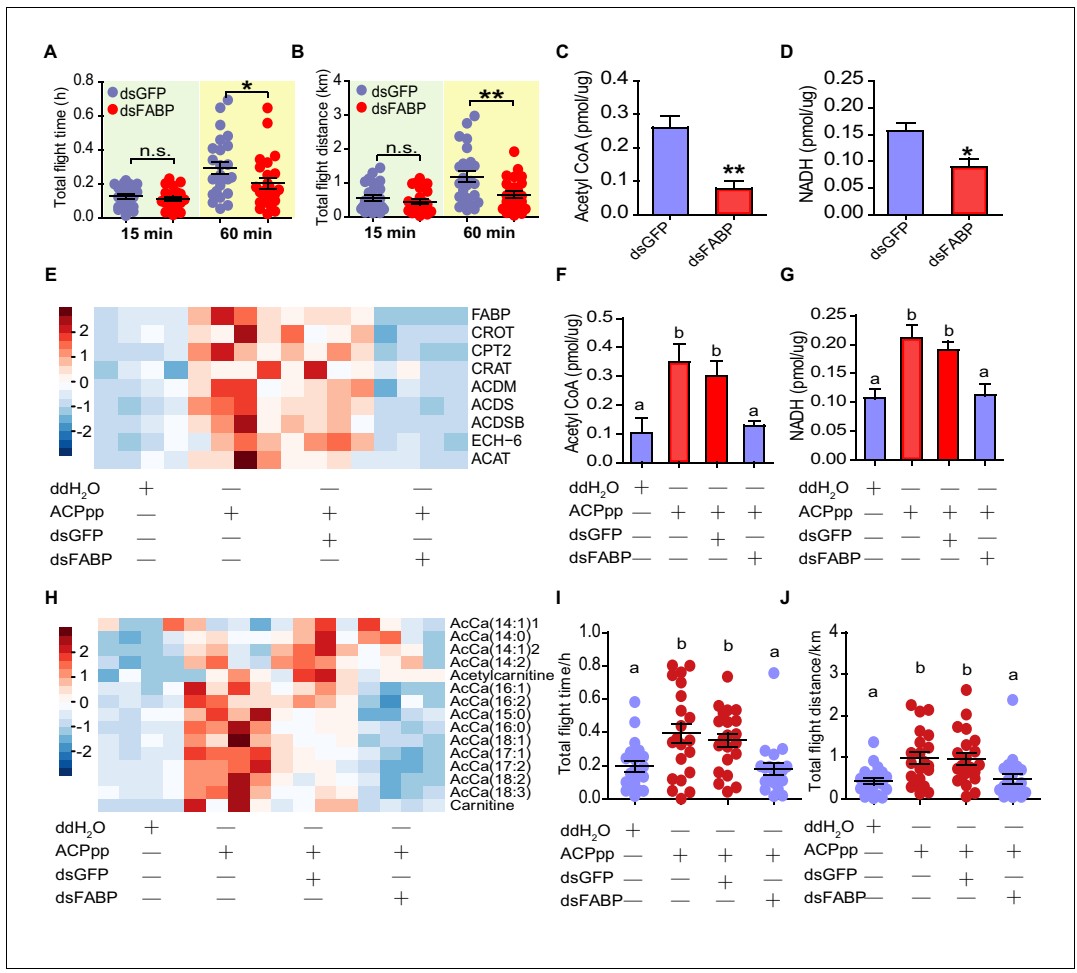

**Figure 6.** FABP mediates the regulatory effects of ACP on lipid metabolism and flight activity. (**A**) Total flight time and (**B**) total flight distance after knockdown of *FABP* in adult locusts (Student's t-test for A, p=0.451, t = 0.7609, df = 44, n = 24 (dsGFP) and 22 (dsFABP) for 15 min, p=0.021, t = 1.992, df = 48, n = 25 (dsGFP) and 25 (dsFABP) for 60 min; for B, p=0.337, t = 0.9713, df = 44, n = 24 (dsGFP) and 22 (dsFABP) for 15 min, p=0.007, t = 2.796, df = 48, n = 25 (dsGFP) and 25 (dsFABP) for 60 min, *p<0.05, **p<0.01). Measurement of (**C**) acetyl CoA and (**D**) NADH in the flight muscle after knockdown of *FABP* in adult locusts (Student's t-test for C, p=0.0011, t = 4.499, df = 10, n = 6 (dsGFP) and 6 (dsFABP), for D, p=0.017, t = 3.124, df = 7, n = 5 (dsGFP) and 4 (dsFABP)). (**E**) Expression levels of beta-oxidation-related genes in flight muscle after *FABP* knockdown in ACP$^{-/-}$ locusts injected with ACP peptide. Unsupervised hierarchical clustering was performed using Clustal 3.0 under uncentered Pearson correlation and average linkage conditions; the results are presented by Java Treeview software. Measurement of (**F**) acetyl CoA, (**G**) NADH, and (**H**) acetyl carnitine in the flight muscle after knockdown of *FABP* in ACP$^{-/-}$ locusts injected with ACP peptide (one-way ANOVA for F, *F* = 7.143, df = 3, p=0.0026, n = 5, 5, 5, and six from left to right, respectively; for G, *F* = 8.673, df = 3, p=0.0012, n = 5, 5, 5, and five from left to right, respectively). Measurement of (**I**) total flight time and (**J**) total flight distance after knockdown of *FABP* in ACP$^{-/-}$ locusts preinjected with ACP peptide (one-way ANOVA for I, *F* = 6.274, df = 3, p=0.0007; for J, *F* = 5.93, df = 3, p=0.001, for both I and J, n = 20, 21, 21, and 20 from left to right, respectively). Different letters indicate that there is a significant difference between two groups. See *Figure 6—source data 1* for details.

The online version of this article includes the following source data and figure supplement(s) for figure 6:

**Source data 1.** Raw data for flight activity, metabolite contents, and gene expression.

**Figure supplement 1.** Flight speed measurement after knockdown of *FABP* in the locust.

## Discussion

The results of this study demonstrated the key roles played by the neuropeptide ACP in the modulation of long-term flight in locusts. ACP peptide is highly abundant in the retrocerebral complex of adult locusts and the *ACP* precursor gene in the brain displays strong transcription responses to

prolonged flight. Mechanistically, ACP facilitates long-term flight by promoting intracellular fatty acid transport to the mitochondria by regulating FABP expression in the flight muscle of locusts (*Figure 7*). Our findings highlight a novel neuroendocrine regulator and help to elucidate relevant mechanism involved in long-term flight in insects.

## Neuropeptidome analysis reveals tissue- and development-specific abundance of neuropeptides in locusts

Through peptidome analysis, we obtained a lot of non-abundant neuropeptides as well as their potential degradation products produced by 37 precursors in the main neuroendocrine tissues, including brain and retrocerebral complex. The neuropeptides detected here including most of previously identified peptides (*Clynen and Schoofs, 2009*). Many of the peptides were abundant at adult stage, indicating their distinct roles in adult-related biology. However, several neuropeptides (e.g. AST-C, inotocin, and AKH4) identified from previous peptidomic study and transcriptome data-based prediction, were not found in the current study. The absence of these neuropeptides in the neuropeptidomic analysis may thanks to their low abundance in tested samples, relative short half-life period, unsuitable chromatographic condition or data acquisition setting. Different sample collection methods as well as multiple mass spectrometry methods may be helpful for systematically identification of all neuropeptides in future.

We also found that the neuropeptides displayed apparently tissue-specific distributions, with either brain- or retrocerebral complex-specific distribution in the locust. By comparing our results with previously peptidome study (*Clynen and Schoofs, 2009*), we found that most of the neuropeptides show similar tissue distribution in the two studies, except for sulfakinin and PVK. The discrepancy between the two studies may be attributed to different sample collection strategies. In the present study, the whole retrocerebral complex of mature adults was used for peptidomic analysis, in contrast to the study of *Clynen and Schoofs, 2009* who analyzed the organs of the retrocerebral complex of immature adults separately.

## ACP has been identified as a novel neuroendocrine player modulating locust flight

Our results suggested that the ACP peptide acts as a neuroendocrine hormone to regulate locust flight capacity. To the best of our knowledge, this study is the first to clearly demonstrate the biological function of ACP in insects. In fact, the ACP peptide was initially isolated from the storage lobes of the CC of migratory locusts and was named locust hypertrehalosemic hormone (Lom-HrTH) because of its activity in the induction of hemolymph trehalose levels in cockroaches but not in locusts (*Siegert, 1999*). Further MS analysis shows that the peptide is highly distributed in CC, hypocerebral ganglion, frontal ganglion, protocerebrum, pars intercerebralis, tritocerebrum, as well as thoracic ganglia of immature Africa migratory locust (*Clynen and Schoofs, 2009*). However, the biological significance of ACP has not been described in detail. Our results show that the *ACP* precursor gene in the brain displays strong transcription responses to prolonged flight. The regulatory roles played by ACP in locust flight are clearly supported by tethered flight experiments after knockdown and knockout of its precursor gene, as well as reduced extended flight ability of ACPR mutants. Based on ours and previous finding,

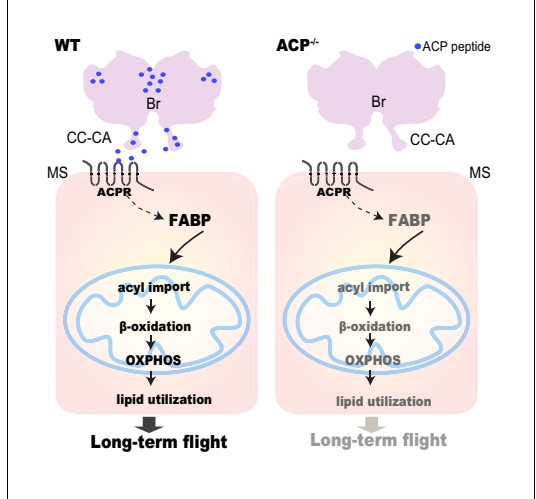

**Figure 7.** Schematic model showing that the neuropeptide. ACP regulates long-term flight by affecting FABP-mediated fatty acid transport and subsequent β-oxidation in the flight muscle. Neuropeptide ACP is produced from the brain and could be secreted into the circulation through CC-CA. Fatty acid transport and lipid utilization are significantly downregulated in the flight muscle of the ACP mutant, thereby resulting in decreased prolonged flight performance.

ACP mature peptide is abundantly detected in the CC of locusts, indicating that ACP may be synthesized in neurosecretory cells of the brain and transported to the storage lobe of the CC via nervi corporis cardiaci I and/or II (*Hekimi and O'Shea, 1987*). Thus, the ACP peptide may be involved in locust flight through secretion into the circulation, to modulate distinct physiological activities in target tissues, such as the flight muscle and fat body, where its receptor is expressed.

The ACP peptide is an insect structural intermediate of corazonin and AKH hormones, all of which belong to the vertebrate gonadotropin-releasing hormone (GnRH) family (*Hansen et al., 2010*). ACP and its cognate receptor have been found in various insects, although they are selectively lost in several insect species, such as the fruit fly (Diptera), the honey bee (Hymenoptera), the pea aphid (Hemiptera), and the body louse (Phthiraptera) (*Hansen et al., 2010*). The role played by ACP in energy metabolism has been suggested in several insect species; for example, this gene has been suggested to regulate hemolymph carbohydrate and lipid levels in *Gryllus bimaculatus* (*Zhou et al., 2018*) and to play a possible role in glycogen hydrolysis in *Platypleura capensis* (*Gäde and Janssens, 1994*). Thus, ACP may play species-specific roles in modifying metabolic activity in insects. Furthermore, it appears that all three GnRH neuropeptides (AKH, ACP, and corazonin) are related to energy embolism to some extent (*Andreatta et al., 2020*), although they have distinct tissue or cellular distributions and do not share overlapping detailed biological roles (*Patel et al., 2014*).

## ACP is involved in the control of lipid transport during long-term flight

Our results demonstrate that ACP modulates long-term flight by primarily affecting lipid transport and utilization in the flight muscle of locusts. This finding is strongly supported by a significant decrease in the levels of genes (e.g. *FABP*, *CPT2*, *CRAT*, *ACDs*, *ECHs*, *HADHs*, *ACATs*) and metabolites (carnitine and acylcarnitine) related to lipid transport and beta-oxidation (*Rubiogozalbo et al., 2004*) in the flight muscle of ACP mutant locusts, as well as the enhanced metabolism-related gene expressions and fatty acid oxidation activity in WT locusts upon ACP peptide administration. Integrating energy metabolism related to long-term flight is a complex and multistep physiological process (*Auerswald and Gäde, 2006*). Generally, the initial flight primarily consumes carbohydrates as an energy substrate, whereas subsequent prolonged flight depends largely on highly efficient lipid utilization as an energy supply (*van der Horst et al., 1993*; *Van der Horst and Rodenburg, 2010*). We show that ACP strongly affects lipid transport and oxidation, not carbohydrate metabolism, in the locust. The changes in lipid metabolism in the flight muscle upon ACP manipulations (both gene knockout and peptide injection) are closely in line with the alteration of long-term flight performance in the parallel treatments, demonstrating the distinct functional roles played by ACP during long-term flight. Despite the inconsistent effects on beta-oxidation products in the flight muscle upon ACP peptide injection (stimulatory effect) and sustained flight (partially inhibitory effect), both two treatments could significantly enhance the expressions levels of gene related to lipid utilization in the flight muscle, implying that the ACP peptide may facilitate lipid utilization in response to prolonged flight. The decreased beta-oxidation products may reflect rapid energy utilization in subsequent mitochondrial metabolism during sustained flight.

## FABP acts as a key molecule for flight-related energy metabolism underlying ACP modulation

We showed that FABP serves as a key molecular target mediating the regulatory effects of ACP on lipid metabolism during locust long-term flight. Although multiple FABP family members are predicted in the locust genome (*Wang et al., 2014*), the FABP identified in this study is specifically expressed in the flight muscle of adult locusts with notably high abundance (*Haunerland et al., 1992*). The mediating role played by FABP in ACP-controlled flight-related lipid metabolism was further confirmed by knockdown and rescue experiments at the molecular, metabolic, and behavioral levels. The significant role of FABP in prolonged flight has also been reported in the desert locust (*Rajapakse et al., 2019*), indicating that FABP plays a conserved role in lipid metabolism and long-term flight in locust species.

FABP has been characterized as an evolutionarily conserved fatty acid carrier that plays essential roles in lipid utilization by affecting mitochondrial beta-oxidation (*Luxon, 1993*; *Luxon et al., 1997*; *Binas and Erol, 2007*). Usually, tissues with high fatty acid oxidative capacities possess more FABP than those that use carbohydrates as an energy source. The locust flight muscle is structurally and

functionally closely related to the mammalian heart muscle, which also depends mostly on fatty acids to fuel its continuous contractions (*Neely and Morgan, 1974*). Therefore, FABP is suggested to be equally important for insect flight muscle and mammalian heart muscle. It has been shown that FABP gene expression in muscle can be upregulated by sustained flying (*Chen and Haunerland, 1994*) and extended physical exercise (*Lammers et al., 2012*). However, the regulatory mechanisms underlying *FABP* transcription by upstream molecules have not been fully characterized. The regulation of FABP expression by ACP during long-term locust flight thus presents a typical case showing the modulation of FABP expression by neuroendocrine factors and may help to elucidate the common molecular mechanisms that participate in the modulation of muscle FABP expression in different species. Further studies are warranted to decipher signaling pathways mediating the regulatory effect of ACP on *FABP* expression, and the results of this research may help to elucidate the precise metabolic mechanisms regulating high energy-demanding activities.

## Multiple neuromodulators are involved in the regulation of flight-related energy metabolism

The central roles played by neuropeptides in glycogen hydrolysis and lipid mobilization have been extensively demonstrated in various insect species (*Arrese and Soulages, 2010*; *Andreatta et al., 2020*). However, few studies have examined the regulatory mechanism underlying flight-related lipid utilization in flight muscle. The transition of substrate utilization from carbohydrates to lipids in the flight muscle has been proposed to be crucial for maintaining long-term flight (*Van der Horst and Rodenburg, 2010*). Glycolysis in flight muscle during early-stage flight is controlled by octopamine (*Mentel et al., 2003*). In comparison, lipid mobilization and glycogen hydrolysis in the fat body during prolonged flight are directly modulated by AKH, whose receptor primarily localizes in the fat body (*Van der Horst, 2003*). The expression levels of all three *AKH* genes strongly increase upon sustained flight in the locust (*Bogerd et al., 1995*). However, enhanced expression after sustained flight was observed only for *AKH2* in our experiments. The difference between the two studies may be attributed to the different sample collection strategy and detection methods used retrocerebral complex for expression analysis examined by qPCR in our study, whereas only CC tissue for expression analysis detected by norther blot in previous work. A strong reduction in lipid metabolites, such as acylcarnitines, acyl-CoA, NADH, triglycerides, and phosphoglycerides, was observed in the flight muscle of ACP mutants, whereas pyruvic acid generated from glycolysis did not change, suggesting that ACP primarily affects lipid metabolism, rather than glycolysis, in the flight muscle of locusts. We infer that these three flight-related regulators have functional differentiation in either a temporally or spatially dependent manner in the modulation of flight metabolism. A coordinated regulatory network involving AKH, ACP, and octopamine is thus proposed to modulate the cooperation of substrate mobilization, transport, and utilization in different tissues during long-term flight. Here, we also revealed that knockdown of either *ACP* or *AKH2* induced similarly suppressed effects on locust flight performance. Further work is warranted to investigate the potential interaction among these neuroendocrine factors in energy regulation associated with flight activity.

Through phenotype examination, we also observed a larger body size in ACP mutants. Usually, the body size of locusts is stable after adult eclosion thanks to its hard exoskeleton. Therefore, the effects of ACP on body size could not be assessed through RNAi of the gene after adult eclosion in the current study. It has been suggested that the growth state for an organism can be negatively affected by other physiological traits, such as locomotion, reproduction, or life span (*Lee et al., 2010*). Therefore, the increased body size of ACP mutants may be attributed to the continuous metabolism changes associated with trade-off effects between flight activity and body growth. Similarly, the loss of function of the AKH peptide results in adult-onset obesity in *Drosophila* (*Gáliková et al., 2015*). These findings may reflect a common role played by ACP and AKH in governing the energy balance of insects. It will be an interesting work to explore the molecular and metabolic basis for body size determination on the basis of established ACP mutant locust line.

In summary, we demonstrate that the ACP peptide acts as a novel neuroendocrine regulator controlling lipid transport and utilization associated with long-term flight in locusts. The ACP-FABP axis involved in long-term flight may serve as an effective molecular target for the prevention of locust plagues and may provide insights into metabolic hemostasis related to sustained locomotion.

# Materials and methods

## Key resources table

| Reagent type (species) or resource | Designation | Source or reference | Identifiers | Additional information |
|---|---|---|---|---|
| Gene (*Locust migratoria*) | ACP | *Yang et al., 2019* | http://www.locustmine.org:8080/locustmine (LOCMI05723) | |
| Gene (*Locust migratoria*) | ACPR | *Yang et al., 2019* | http://www.locustmine.org:8080/locustmine (comp340809_c0_seq1) | |
| Genetic reagent (*Locust migratoria*) | GeneArt Precision gRNA Synthesis Kit | ThermoFisher | A29377 | |
| Genetic reagent (*Locust migratoria*) | Cas9 protein | Invitrogen | A36496 | |
| Antibody | (anti-ACP rabbit polyclonal) | This paper | Produced by ABclone, China | (1:200) |
| Antibody | Alexa Fluor-488 goat anti-rabbit IgG | Life Technologies | Cat. A-11008 | (1:500) |
| Antibody | anti-FABP | This paper | Developed by ABclone, China | (1:5000) |
| Antibody | anti-CPT2 | Abcam | ab153869 | (1:1000) |
| Antibody | anti-ACDM | Abcam | ab92461 | (1:1000) |
| Antibody | Goat anti-rabbit IgG secondary antibody | EASYBIO | BE0101-100 | (1:5000) |
| Antibody | Polyclonal antibody against tubulin | This paper | Produced by ABclone, China | (1:5000) |
| Peptide, recombinant protein | ACP peptide | ABclone | pQVTFSRDWSP Gamide | |
| Commercial assay or kit | Acetyl-CoA assay kit | Sigma-Aldrich | MAK039 | |
| Commercial assay or kit | NAD/NADH Assay Kit | Abcam | ab65348 | |
| Commercial assay or kit | Pyruvate Colorimetric/ Fluorometric Assay Kit | BioVison | K609 | |
| Commercial assay or kit | Citric acid content detection kit | Solarbio | BC2150 | |
| Chemical compound, drug | LightCycler 480 SYBR Green I Master | Roche | 04887352001 | |
| Software, algorithm | GraphPad Prism 5 | GraphPad Software | RRID:SCR_002798 | |

## Insect rearing

Locusts used in these experiments were obtained from a colony maintained at the Institute of Zoology, Chinese Academy of Sciences, Beijing, China. Both nymphs (300–400 insects per cage) and adult locusts (~100 per cage) were reared under a 14:10 light:dark photocycle regime at 30 ± 2°C. The locusts were fed with fresh wheat seedlings and bran (*Hou et al., 2017*). For age definition,

adult locusts between 0 and 12 hafter molting were referred as day 0 post-adult eclosion (PAE 0 day).

## Quantitative neuropeptidome analysis

For sample preparation, the brain tissues containing only the protocerebrum, deuterocerebrum, and tritocerebrum and the retrocerebral complex (including corpora cardiaca, corpora allata, hypocerebral ganglion, and small neuronal structures) from 35 fifth-instar female nymphs (the third day after molting) and 35 mature female adults (PAE 10 days) were carefully microdissected and frozen in liquid nitrogen. All samples were stored at −80°C. Three independent biological replicates were prepared for each sample.

Tissues were homogenized in 200 μl lysis buffer (methanol/ddH$_2$O/formic acid = 90/9/1) through sonication on ice followed by centrifugation at 12,000 × rpm for 20 min at 4°C to remove insoluble fractions. The supernatants were ultrafiltered with a 10 kDa ultrafiltration column and freeze-dried for peptide collection. The peptide pellets were resuspended in 20 μl 0.1% formic acid prior to MS analysis (*Han et al., 2015*). LC-MS/MS analysis was performed on an Easy-nLC 1000 (Thermo Fisher Scientific, Bremen, Germany) coupled LTQ-Orbitrap Elite (Thermo Fisher Scientific) hybrid mass spectrometer. Peptides were separated on a column packed with 2 μm C18 (100 Å, 75 μm x 50 cm, Thermo Fisher Scientific) using a 130 min gradient from 3–30% acetonitrile (0.1% formic acid) with a flow of 250 nL/min. The eluted neuropeptides were injected into the mass spectrometer via a nano-ESI source (Thermo Fisher Scientific). Ion signals were collected in a data-dependent mode and run with the following settings: full scan resolution at 70,000, automatic gain control (AGC) target 3E6; maximum inject time (MIT) 20 ms; scan range m/z 300–1800; MS/MS scans resolution at 17,500; AGC target 1E5; MIT 60 ms; isolation window 2 m/z; normalized collision energy 27; loop count 10; charge exclusion: unassigned, 1, 8,>8; peptide match: preferred; exclude isotopes: on; dynamic exclusion: 30 s; dynamic exclusion with a repeated count: 1. The MS/MS data were acquired in raw files using Xcalibur software (version 2.2, Thermo Fisher Scientific).

The extracted MS/MS spectra were searched against a composite database of *Locust migratoria* (3286 protein sequences, download from NCBI, 2019) and a protein database (containing 17,307 protein sequences, http://www.locustmine.org:8080/locustmine) (*Yang et al., 2019*) using in-house PEAKS software (version 7.0, Bioinformatics Solutions, Waterloo, Canada). The database search parameters: parent ion mass tolerance is 15 ppm, and fragment ion mass tolerance is 0.05 Da; enzyme specificity, none. The following modifications were applied: C-terminal amidation (A, −0.98) and pyroglutamination from Q (P, −17.03), maximum missed cleavages per peptide: 2, and maximum allowed variable PTM per peptide: 2. A fusion target and decoy approach was used to for the estimation of the false discovery rate (FDR) and controlled at ≤1.0% at the peptide level. Identified neuropeptides were further validated by comparison with predicted neuropeptide precursors and previous neuropeptidome analysis in the locust (*Clynen and Schoofs, 2009*; *Hou et al., 2015*).

Relative quantification of the neuropeptidome was performed by the label-free approach in the PEAKS Q module (*Han et al., 2015*). Feature detection was performed separately on each sample by using the expectation-maximization algorithm. The features of the same peptide from different samples were reliably aligned together using a high-performance retention time alignment algorithm. The identification results were chosen to attach as the last step of the label-free quantification. Peptides were determined to be significantly changed between different samples if Student's t-test yielded p-values<0.01.

## RNA extraction and qPCR

Total RNA of experimental samples was isolated using TRIzol reagent (Invitrogen). RNA quantification and reverse transcription were performed as previously described (*Hou et al., 2017*). Transcript levels of target genes were detected by SYBR Green kit on a LightCycler 480 instrument according to the manufacturer's instructions (Roche). *RP49* was used as an internal reference. Dissociation curves were determined for each gene to confirm unique amplification. At least four biological replicates were performed for gene expression level analysis. The primers are shown in *Supplementary file 4*.

## Tissue-specific expression pattern analysis

Tissues including the brain containing only the protocerebrum, deuterocerebrum, and tritocerebrum, thoracic ganglion, retrocerebral complex, fat body, flight muscle, and ovary (or testis) of adult locusts at PAE 7 days were dissected and frozen in liquid nitrogen. All samples were stored at −80°C. Four independent biological replicates were prepared for each sample. Tissues of six to eight individuals were collected for each replicate. RNA extraction and qPCR analysis were performed as described above.

## RNA interference

Double-stranded RNAs of target genes were synthesized using the T7 RiboMAXTM Expression RNAi system (Promega, USA). The dsRNA was first injected into the hemolymph of adult locusts at day one post-adult eclosion (PAE 1 day, 6 µg/locust). A second injection was performed at PAE 4 days. dsGFP was used as the control. Flight performance was measured 3 days after the second injection (PAE 7 days).

## Tethered flight assay

The flight activities of individual locusts were measured by using a computer-aided flight-mill device modified from previous studies (Beerwinkle et al., 1995). The arm length of the fight-mill is 12 cm, and the flight circumference is 0.75 m. One plastic rod forms the upper, free-turning body of the flight mill that supports the mill arm and serves as a bearing on the pivot pin. The locust was tethered at the end of the lightweight arm, which allowed it to fly along the cycle derived from the arm. The interruption of an infrared beam by a rotating arm generated an electrical signal that was recorded by the computer. Flight was induced by a fan placed above the flight mill (blow for 2 s with an interval of 30 s, 1.5 m/s wind speed). During the assay, four flight parameters were obtained for each locust, including total flight distance and duration and average flight velocity and maximum flight velocity. The flight mill device was positioned in a room under a photoperiod of 14:10 (L:D) h at 30 ± 1°C. Locusts that did not flight in this assay were excluded. At least 20 locusts were used in each treatment. Individuals were randomly allocated into experimental group and control group, and no restricted randomization was applied.

For sample preparation of flight treatment, the locusts were forced to sustain flight for 15, 30, 45, and 60 min, respectively. Insects that stopped flying were artificially stimulated to continue flight. Brains of tested locusts were collected, rapidly frozen in liquid nitrogen and stored at −80°C. Four independent biological replicates were prepared for each time point. Tissues of six to eight individuals were collected for each replicate. Individuals who could not finish sustained flight were discarded.

## Generation of ACP and ACPR mutant locusts using the CRISPR/Cas9 system

The establishment of mutant locusts using the CRISPR/Cas9 system was performed as previously described (Li et al., 2016). The gRNAs containing 20 bases adjacent to a PAM sequence for both ACP and ACPR were designed using the CasOT tool. The gRNAs were synthesized using the GeneArt Precision gRNA Synthesis Kit (ThermoFisher, A29377). In brief, a 13.8 nl mixture of purified Cas9 protein (Invitrogen, A36496, Massachusetts, USA) and guide RNA of target genes (final concentrations: 300 and 150 ng/µl, respectively) was injected into the newly collected embryos (2 hr after production) using a microinjector. The injected embryos were then placed in a 30°C incubator until the nymphs hatched. The hatched nymphs were reared as described above. For genotype analysis, part of the middle foot of each adult locust was collected and lysed with 45 µl NAOH buffer (50 mM) at 95°C for 30 min and then neutralized by adding 5 µl Tris-HCl (1 M, pH = 8.0). The supernatant (3 µl) was used as the PCR template to amplify the targeted DNA fragment. Primers for gRNA synthesis were shown in Supplementary file 4.

## Whole-mount immunohistochemistry assay

The brains of adult locusts were fixed in 4% paraformaldehyde overnight. After being washed with 1 × PBS buffer, the samples were blocked with 5% BSA for 1 h and then incubated with affinity-purified polyclonal rabbit antibody against ACP (produced by ABclone, China, 1: 200) at 4°C for 24 h.

Alexa Fluor-488 goat anti-rabbit IgG (Cat. A-11008, 1: 500; Life Technologies) was used as the secondary antibody. Fluorescence was detected using an LSM 710 confocal laser-scanning microscope (Zeiss). Negative controls were imaged under the same detection conditions as positive staining.

## Survival rate measurement

Adults of WT and ACP$^{-/-}$ locusts at PAE 1 day were reared under a 14:10 light/dark cycle at 30℃. The numbers of dead insects were assessed every day. The survival curves were drawn using GraphPad Prism five software. Differences in the survival rate of females or males between WT and ACP mutant locusts were compared by using the log-rank (Mantel-Cox) test method. At least 50 individuals were assayed in each group to determine the survival rate.

## Phylogenetic tree construction

The protein sequences of ACPR from *Tribolium castaneum* and *Aedes aegypti* were used as seed sequences to search their homologs in the locust genome and transcriptome database using the tblastn algorithm. Three kinds of structure-related neuropeptide receptor proteins, adipokinetic hormone receptor (AKHR), ACPR, and corazonin receptor (CRZR), from locusts and several representative insect species were used to construct their phylogenetic relationship by using MEGA software (*Tamura et al., 2011*).

## Western blot analysis

Total proteins from flight muscles of WT and ACP$^{-/-}$ locusts were extracted using TRIzol reagent, as previously described (*Hou et al., 2019*). The protein extracts (50 µg) were electrophoresed on 4–20% Biofuraw precast gels (Tanon, China) and then transferred to polyvinylidene difluoride (PVDF) membranes (Millipore). The membrane was incubated with polyclonal antibody against target protein (anti-FABP, developed by ABclone, China, 1:5000; anti-CPT2, Abcam, 1:1000; anti-ACDM, Abcam, 1:1000). Goat anti-rabbit IgG (EASYBIO, 1:5000) was used as the secondary antibody. Polyclonal antibody against tubulin was used as an internal control. Protein bands were detected by chemiluminescence (ECL kit, Thermo Scientific).

## RNA-seq and analysis

The flight muscle and fat body tissues were dissected from WT and ACP$^{-/-}$ female locusts under resting state at PAE 7 days. Total RNA of these samples with three biological replicates was extracted using TRIzol (Invitrogen) and treated with DNase I following the manufacturer's instructions. RNA quality was assessed using an Agilent 2100 Bioanalyzer (Agilent) to verify RNA integrity. cDNA libraries were prepared according to Illumina's protocols. The adaptor sequences in the raw sequencing data were filtered using Trimmomatic-0.30. Clean reads were mapped to the locust genome sequence using Tophat software. The number of total reads was normalized by multiple normalization factors. Transcript levels were calculated using the reads per kb million mapped (RPKM) reads criteria. The differences between the test and control groups were based on *P* values with false discovery rate (FDR) correction. Differentially expressed genes with FDR < 0.1, Log$_2$ (FC) >1, and RPKM > 0.5 in each comparison were enriched. The raw sequence data reported in this paper have been deposited in the Genome Sequence Archive (Genomics, Proteomics & Bioinformatics 2017) in National Genomics Data Center, Beijing Institute of Genomics (China National Center for Bioinformation), Chinese Academy of Sciences, under accession number CRA003348 that are publicly accessible at https://bigd.big.ac.cn/gsa.

## Metabolome analysis

The flight muscle was dissected from WT and ACP$^{-/-}$ female locusts under resting state at PAE 7 days. The metabolomic profile analysis contained extraction, separation, and detection of metabolites together with metabolomic data processing. For metabolite extraction, 20 mg flight muscle was homogenized in 600 µl cold extraction buffer (V$_{methanol}$/V$_{ddH2O}$, 8/2). The 300 µl homogenate was transferred to a new tube containing 900 µl methyl tert-butyl ether (MTBE) and 250 µl ddH$_2$O. After 10 min of shaking, the mixture was centrifuged at 13,000 × g at 4℃ for 10 min. Lipid extract in the upper layer and polar metabolites in the lower layer were collected for further freeze drying. Lipid

pellets and polar metabolites were thus resuspended using acetonitrile/isopropanol and acetonitrile/ddH$_2$O for subsequent chromatographic separation, respectively.

For polar extract separation, untargeted metabolomics analysis was conducted on an Ultimate 3000 ultra-high-performance liquid chromatograph coupled with a Q Exactive quadrupole-Orbitrap high-resolution mass spectrometer UPLC-HRMS system (Thermo Scientific, USA). The polar metabolome extracts were profiled on reversed-phase chromatographic separation with positive and negative ionization detection, respectively. Metabolites were separated by using an Acquity HSS C18 column (Waters Co., USA, 2.1 × 100 mm) and eluted by 0.1% formate/water and acetonitrile using linear gradient ramping from 2% organic mobile phase to 98% in 10 min. Furthermore, other mobile phases consisting of water and ammonium acetonitrile/methanol both containing ammonium bicarbonate buffer salt were employed to elute metabolites separated on an AcquityTM BEH C18 column (Waters Co., USA, 1.7 μm, 2.1 × 100 mm), the gradient was used as follows: 0 min 2% organic phase ramped to 100% in 10 min, and another 5 min was used for column washing and equilibrating. The flow rate, injection volume and column temperature were all set at the same conditions, 0.4 ml/min, 5 μl and 50℃, respectively.

## MS of polar extracts

The quadrupole-Orbitrap mass spectrometer was operated under identical ionization parameters with a heated electrospray ionization source except ionization voltage including sheath gas 45 arb, aux gas 10 arb, heater temperature 355℃, capillary temperature 320℃ and S-Lens RF level 55%. The metabolome extracts were profiled with full scan mode under 70,000 FWHM resolution with AGC 1E6 and 200 ms max injection time. A 70 ~ 1000 m/z scan range was acquired. QC samples were repeatedly injected into the acquired Top 10 data-dependent MS2 spectra (full scan-ddMS2) for comprehensive metabolite and lipid structural annotation. The 17,500 FWHM resolution settings were used for full MS/MS data acquisition. Apex trigger, dynamic exclusion and isotope exclusion were turned on, and the precursor isolation window was set at 1.0 Da. Stepped normalized collision energy was employed for collision-induced disassociation of metabolites using ultrapure nitrogen as the fragmentation gas. All the data were acquired in profile format.

## Separation of lipid extraction

The chromatographic separation of untargeted lipidomics was performed under positive and negative ionization detection modes, respectively, as described above. An Accucore C30 core-shell column was utilized for lipid molecule separation at 50℃, which was eluted with 60% acetonitrile in water (A) and 10% acetonitrile in isopropanol (B), with both containing 10 mM ammonium formate and 0.1% formate. The separation gradient was optimized as follows: initial 10% B ramping to 50% in 5 min and further increasing to 100% in 23 min, the other 7 min for column washing and equilibration using 0.3 mL/min flowrate.

The ionized lipid molecules were detected using the same parameters as previously described. Lipid extracts (300–2000 m/z) were profiled with the same parameters as the metabolome used. Lipids were structurally identified by acquiring data-dependent MS2 spectra, and the key settings included 70,000 FWHM full scan resolution, 17,500 FWHM MS/MS resolution, loop count 10, AGC target 3e6, maximum injection time 200 ms and 80 ms for full scan and MS/MS, respectively, and dynamic exclusion 8 s. Stepped normalized collision energy 25% + 40% and 35% were employed for positive and negative mode after optimization.

## Data processing

The full scan and data-dependent MS2 metabolic profile data were further processed with Compound Discoverer software for comprehensive component extraction. The polar metabolites were structurally annotated by searching acquired MS2 against a local proprietary iPhenome SMOL high resolution. The MS/MS spectrum library was created using authentic standards, the NIST 17 Tandem MS/MS library (National Institute of Standards and Technology), the local version MoNA (MassBank of North America), and the mzCloud library (Thermo Scientific, USA). In addition, the exact m/z of MS1 spectra was searched against a local KEGG and HMDB metabolite chemical database. For metabolite identification or structural annotation, the mass accuracy of the precursor within ±5 ppm was a prerequisite; meanwhile, isotopic information including at least 1 isotope within 10 ppm and a

fit score of a relative isotopic abundance pattern of 70% were introduced to confirm the chemical formula in addition to the exact mass. Furthermore, retention time information as well as high resolution MS/MS spectra similarity was employed to strictly confirm the structural annotation of metabolites. The area under curve values were extracted as quantitative information of metabolites with XCalibur Quan Browser information, and all peak area data for the annotated metabolites were exported into Excel software for trimming and organization before statistical analysis (Microsoft, USA). On the other hand, untargeted lipidomics data were processed with LipidSearch software, including peak picking and lipid identification. The acquired MS$^2$ spectra were searched against in silico predicted spectra of various compounds, including phospholipids, neutral glycerolipids, spingolipids, neutral glycosphingolipids, glycosphingolipids, steroids, and fatty esters. The mass accuracies for the precursor and MS/MS product ion searches were 5 ppm and 5 mDa, respectively. The MS/MS similarity score threshold was set at 5. The potential ionization adducts include hydrogen, sodium, and ammonium for positive ion and hydrogen loss, as well as formate and acetate adducts for negative mode. Lipid identification was strictly manually checked and investigated one-by-one to eliminate false positives chiefly based on peak shaking, adduct ion behavior, fragmentation pattern, and chromatographic behavior.

## Data statistics

The metabolome and lipidome data derived from different measurements were normalized to the sample weight used prior to further processing. Then, the resultant quantitative information from the foregoing methods was merged, and those detected with multiple methods were excluded to guarantee the uniqueness of metabolites and lipids. Log10 was then transformed for final statistical analysis. Principal component analysis was conducted with SIMCA-P software (Umetrics, Sweden), and other univariate analyses, including independent sample t-test and p value FDR adjustment, as well as metabolic pathway analysis, were conducted on the MetaboAnalyst website.

## Measurement of metabolites

For measurement of acetyl-CoA, NADH, pyruvate, and citric acid, tissues (20–40 mg) of 3–4 individuals were dissected out and homogenized in distinct extraction buffer. The extraction solution was centrifuged at 10,000 × g at 4℃ for 10 min. Of these supernatants, 6 μl was aspirated for protein determination, and the rest was deproteinized with a 10 kDa molecular weight cutoff spin filter to remove proteins prior to the reaction according to the manufacturer's procedures. Intracellular acetyl-CoA levels were measured by the acetyl-CoA assay kit (Sigma, MAK039) following the manufacturer's instructions. A 50 μl sample was added to the reaction mixture and incubated for 10 min at 37℃. Fluorescence intensity (λexcitation = 535/λ emission = 587 nm) was measured for acetyl-CoA detection. The NADH content was measured using the NAD/NADH Assay Kit (Abcam, ab65348). Samples were incubated at 60℃ for 30 min to remove NAD$^+$. The 20 μl sample was added to the reaction mixture and incubated for 1 h- at room temperature. The absorbance at 450 nm was measured for NADH detection. Pyruvate was detected by using the Pyruvate Colorimetric/Fluorometric Assay Kit (BioVison, K609) according to the manufacturer's instructions. The 20 μl sample was added into the reaction mix and incubated at room temperature for 30 min. The absorbance at 570 nm was measured in a microplate reader. Citric acid was detected using the citric acid content detection kit (Solarbio, BC2150) following the manufacturer's instructions. A 50 μl sample was used in the reaction. The absorbance at $OD_{545}$ nm was detected for citric acid measurement using SpectraMax Plus 384. For all metabolite measurements, the background was corrected by subtracting the blank standard value from all readings. All data were normalized to the protein concentration, which was measured by using the BCA method. At least five biological replicates were performed for each treatment.

## Peptide injection assay

For both WT locusts and ACP$^{-/-}$ mutants, commercially synthetic ACP peptide (20 pmol, 2 μL, ABclone) was injected into the hemolymph of female ACP$^{-/-}$ adults every two days beginning at PAE 1 day. To validate the involvement of FABP in the regulation of energy metabolism and flight activity by ACP peptide, metabolites in muscles and flight performance were measured in ACP$^{-/-}$ locusts injected with ACP peptide or combined with injection of ACP peptide and ds*FABP*. For ACP

peptide and ds*FABP* dual treatments, ds*FABP* (6 µg/µL) was mixed with ACP peptide (40 pmol) and injected into the ACP$^{-/-}$ adults at PAE 3 days. Tissue collection and flight performances of tested insects were both conducted at PAE 7 days.

## Measurement of acyl carnitines

For measurement of acyl carnitine, muscle tissues (20 ± 0.5 mg) were homogenized in 600 µL pre-chilled methanol-water (8/2, v:v) solution containing deuterium-labeled internal standards (200 ng/mL AcCa(16:0)-D3, 20 ng/mL AcCa(12:0)-D3, 10 ng/mL AcCa(8:0)-D3, 200 ng/mL AcCa(2:0)-D3, 50 ng/mL Carnitine-D9). The homogenates were vortexed and centrifuged at 13,000 × g for 10 min at 4°C. 250 µL aliquots of supernatant were transferred into another EP tubes and dried in vacuum using the CentriVap Concentration Systems (Labconco Corporation, USA). The extracts were dissolved in 150 µL methanol-water (5:5, v:v) for further chromatography analysis.

## Statistical analysis

Statistical methods for omic analysis were performed as described above. The data that do not meet normal distribution was excluded for the analysis of gene expression, biochemistry assay, and flight activity. All data are presented as the mean ± SEM and statistically analyzed using GraphPad Prism five software. Two-tailed unpaired student's t-test was used for two-group comparisons, and one-way ANOVA followed by Tukey's post hoc test was used for multigroup comparisons. Differences were considered statistically significant at $p < 0.05$.

## Acknowledgements

We thank Weichan Cui for helpful work in IHC experiment, Liya Wei for sample collection of peptidome experiment, Bin Han for neuropeptidome analysis, and Baozhen Du for helpful discussion on this work. This study was supported by the National Natural Science Foundation of China (Grant NO. 31930012 and 32070497) and grants from Chinese Academy of Sciences (nos. 152111KYSB20180036) and Youth Innovation Promotion Association CAS (No. 2021079).

## Additional information

### Funding

| Funder | Grant reference number | Author |
|---|---|---|
| National Natural Science Foundation of China | 31930012 | Xianhui Wang |
| National Natural Science Foundation of China | 32070497 | Li Hou |
| Chinese Academy of Sciences | 152111KYSB20180036 | Le Kang |
| Youth Innovation Promotion Association of the Chinese Academy of Sciences | 2021079 | Li Hou |

The funders had no role in study design, data collection and interpretation, or the decision to submit the work for publication.

### Author contributions

Li Hou, Conceptualization, Data curation, Funding acquisition, Methodology, Writing - original draft; Siyuan Guo, Software, data analysis; Yuanyuan Wang, Ding Ding, Data curation, Methodology; Xin Nie, Beibei Li, Methodology; Pengcheng Yang, Data curation, data analysis; Le Kang, Conceptualization, Funding acquisition, Writing - review and editing; Xianhui Wang, Conceptualization, Funding acquisition, Writing - original draft, Writing - review and editing

## Author ORCIDs

Le Kang (ID) https://orcid.org/0000-0003-4262-2329
Xianhui Wang (ID) https://orcid.org/0000-0002-8732-829X

## Decision letter and Author response

Decision letter https://doi.org/10.7554/eLife.65279.sa1
Author response https://doi.org/10.7554/eLife.65279.sa2

## Additional files

### Supplementary files

- Supplementary file 1. Neuropeptides identified in the brain and retrocerebral complex by peptidome.
- Supplementary file 2. Mutation efficiency of the *ACP* gene in G0 and G1 generation locusts.
- Supplementary file 3. Mutation efficiency of the *ACPR* gene in G0 and G1 generation locusts.
- Supplementary file 4. Primers for qPCR, gRNA synthesis, and RNAi experiment.
- Transparent reporting form

### Data availability

All data generated or analysed during this study are included in the manuscript and supporting files. Source data files have been provided for Figures 1, 2, 3, 4, 5, and 6.

The following dataset was generated:

| Author(s) | Year | Dataset title | Dataset URL | Database and Identifier |
|---|---|---|---|---|
| Yang P | 2020 | RNA-Seq of fat body and muscle tissues with mutant ACP neuropeptide | https://ngdc.cncb.ac.cn/gsa/browse/CRA003348 | National Genomics Data Center, gsa |

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
