## [Decision Letter]

**Acceptance summary:**

This study provides an insight into the rewiring of metabolism in insect muscles to support the energy requirements of long-range flight. It demonstrates the molecular mechanisms by which signaling neuropeptides released by the insect brain act via cell-surface receptors on muscle cells to regulate metabolism during long-range flight.

**Decision letter after peer review:**

Thank you for submitting your article "Neuropeptide ACP facilitates lipid oxidation and utilization during long-term flight in locusts" for consideration by *eLife*. Your article has been reviewed by 3 peer reviewers, including Raghu Padinjat as the Reviewing Editor and Reviewer #1, and the evaluation has been overseen by K VijayRaghavan as the Senior Editor. The following individual involved in review of your submission has agreed to reveal their identity: Aravind Ramanathan (Reviewer #2).

Summary:

This paper addresses the metabolic adaptations required to sustain long term flight in insects and the potential of the neuropeptide ACP to regulate these processes. This is a scientific problem which is interesting and important for a metabolic and neuroendocrinology perspective. The paper is of general importance because locusts are important pest insects. This study is expected to have an important impact for understanding how locusts (and numerous other animals) modulate muscle metabolism in response to physiological needs. The study is done elegantly and provides interesting new insights into the above process.

Essential Revisions:

1) While a potential ACP receptor is presented on the basis of informatics and expression analysis, there is no functional evidence of a role for this protein in mediating the actions of ACP. The authors can show this in vitro by expressing the receptor in a cellular expression system and challenge the receptor with synthetic ACP.

and/or to substantiate the mechanism of action of ACP in vivo, the authors should compare the phenotype of the RNAi-knockdown of the ACP receptor ( or Crispr-Cas9 knockout) with the flight phenotypes of WT and FABP knockdowns.

2) A key strategy in this study is to compare phenotypes, transcriptomes and metabolomes between wild type and ACP-/- animals. A body size defect is reported for ACP-/- locusts that is also seen with the RNAi. This is an important point as in insects adult body size is determined during development and will also imply effects of metabolism. In general, studying rapidly adaptive processes like metabolism (that change based on seasons and nutritional availability) using knockout lines can lead to alternative metabolic adaptations that will be difficult to map. This is a concern in an otherwise elegant study.

The authors should strengthen the conclusion that enhanced ACP activity in adult locusts is responsible for the reported transcriptome and metabolome changes leading long flight modulation. Can they provide data that:

a) Please show if there is altered staining of ACP in the brain during long flight conditions ?

b) Does the increased ACP expression actually modulate fatty acid oxidation in response to extended flight as suggested. Can injection of ACP peptide alter FAO capacity and map to extended flight capability?

c) If the changes in transcriptome and metabolome seen in ACP-/- are a direct reflection of the function of this peptide is modulating metabolism in long-flight, then are the changes in these transcripts and metabolites also seen in wild type locust tissue when comparing resting with tissue after 1 hr flight ?

3) Clarity on methodology

Please provide clarity on some methodology issues to make the work more informative and accessible. The following points to be addressed:

i) Hou et al. analysed the brain and CC-CA neuropeptidome. For this purpose, they prepared samples of brains and CC-CA. To be able to understand the tissue distribution of the neuropeptides, it is important to know what is meant by 'brain' samples and by CC-CA samples. Does the brain also contain the optic lobes, frontal ganglion, suboesophageal ganglion, the NCC or only the proto-, deutero-, and tritocerebrum? Do the CC-CA samples only contain the CC and the CA or also the hypocerebral ganglion and other small neuronal structures that are closely attached to the CC-CA?

ii) Although the neuropeptidomics / mass spectrometry data do not directly influence the conclusions of the manuscript, it is crucial that the authors clarify the trypsin setting in their MS analysis. Because this may be a major mistake. It is hard to believe that the neuropeptidomics data were generated by this method. Explanation: Trypsin generates theoretical peptide spectra that end in R or K. So it is impossible that these theoretical tryptic peptide spectra were used for the identification of naturally occurring neuropeptide spectra (as these do not by definition end in R or K).

iii) Please revise the annotation of the peptidomic data in line with the conventions or existing literature for the field so that this work can be used in conjunction with existing literature. The authors should compare their data with the existing neuropeptidomic tissue distribution literature, for e.g in Clynen and Schoofs, (2009). (detailed comments on this point provided in the individual reviews below)

iv) It is not clear if the transcriptome analysis was performed on resting muscle tissue or on tissue collected after 1 hr flight. Same applies to metabolomics analysis.

v) In all figure notations such as a, ab, b, bc , abc have been used. It is apparently related to statistical analysis but this is very confusing and not well explained. The statistical analysis in all figure should be explained much more clearly.

*Reviewer #1:*

To identify neuropeptides that might regulate muscle metabolism during sustained flight, the authors carried out mass spectrometry to identify peptides that are expressed in the brain and neuroendocrine tissues using high-resolution mass spectrometry. From the peptides found in these tissues, they shortlisted those (ACP, AKH2 and NPF1) whose levels were elevated during long-term flight; of these, when depleted by RNAi, ACP and AKH2. A role for AKH2 in controlling metabolism and flight has been previously described. Here the authors study the role of ACP in this process.

To study the role of ACP the authors deleted the gene using CRISPR/Cas9 gene editing and demonstrate loss of the peptide in edited animals by immunohistochemistry. Knockout animals were larger than controls and showed shorter flight times and distances travelled. This suggests a role for ACP in regulating long term flight. Using bioinformatics analysis, the authors identify the likely candidate receptor for ACP (ACPR) that could mediate the actions of ACP. It is highly expressed in muscle and fat body. However the functional significance of this receptors in terms of ACP action remains to be established.

To understand the mechanism by which ACP supports flight a transcriptome analysis was carried out on flight muscle and fat body tissue comparing wild type and ACP-/-. From a statistical analysis the authors conclude that the likely target is muscle although this remains to be established. Pathway analysis of transcript changes indicate alteration in molecules involved in metabolism. Perhaps an analysis of transcriptional changes in wild type muscle will help to identify transcript changes relevant to adaptation to long term flight. A key molecule identified in this analysis was FABP, a key transporter of fatty acids. The functional importance of this molecule was determined by the RNAi depletion of FABP in muscle which resulted in defects in a tethered flight assay.

To determine the impact of these transcriptional changes ACP-/- muscles on metabolism, a metabolome analysis was performed on muscle tissue using LC-MS. A key finding was the downregulation of several molecular species of acylcarnitines with no change in the levels of pyruvate, the end product of glycolysis. Together with the transcriptome analysis, this study concludes that ACP signalling during long flight remodels metabolism to enhance mitochondrial metabolism.

The definition of long-term flight as used in the context of this study and the phenomenon being discussed have not been clearly defined. This should be done early on in the manuscript.

While the manuscript makes a case for the function of ACP, what is the role of AKH2. Clearly not sufficient to account for loss of ACP. What is the phenotype of a depletion of both AKH2 and ACP. AKH2 Can RNAi for AKH2 be done in ACP-/- animals and if so what would be the flight defect ?

A body size defect is reported for ACP-/-. Is this also seen with the RNAi. This is an important point as in insects adult body size is determined during development and will also imply effects of metabolism. Please clarify this point.

What is the effect of RNAi depletion, in muscle, of ACPR on flight duration and length. Can it phenocopy the ACP depletion or knockdown ? At the moment this data is of limited value.

It is not clear if the transcriptome analysis was performed on resting muscle tissue or on tissue collected after 1 hr flight. Same applies to metabolomics analysis.

Also what are the transcriptome changes in wild type muscle tissue comparing resting and 1 hr flight. This information will help identify the changes relevant to long term flight in the ACP-/- and separate out transcriptional changes that are developmental compared to wild type.

In all figure notations such as a, ab, b, bc , abc have been used. It is apparently related to statistical analysis but this is very confusing. The statistical analysis in all figures should be explained much more clearly.

*Reviewer #2:*

– The authors aim to understand how locusts modulate skeletal muscle metabolism to support long flights that is a well known aspect of the animal's natural behavior. The authors hypothesize that a neuropeptide directly signals to flight muscles and promote a metabolic switch (fatty acid β oxidation) that can support long flights. The authors aimed to identify one or more neuropeptides that can perform this metabolic switch.

– In order to identify the neuropeptide the authors used a combination of mass spectrometry based peptidomics, mRNA profiling and RNAi mediated knockdown studies and immunofluorescence staining. This part of the study identified 2 neuropeptides ACP and AKH2 that were upregulated in response to sustained flight. Since ACP was less studied in the context of this study, the authors proceeded with this peptide for functional studies.

– Using CRISPR knockout lines, the authors showed the functional dependency of this peptide on sustained flight.

– They identified the role of this peptide in promoting β oxidation of fatty acids (using metabolomics and mRNA profiling) as judged by impaired β oxidation in ACP knockout animals. This effect was recued by peptide treatment.

– The authors next showed a specific FABP protein as a mediator of the effects of ACP in promoting fatty acid transport and β oxidation in response to this neuropeptide.

The systems analysis of metabolism is a major strength of this paper.

In general, studying rapidly adaptive processes like metabolism (that change based on seasons and nutritional availability) using knockout lines can lead to alternative metabolic adaptations that will be difficult to map. This is a concern in an otherwise elegant study. Though not highlighted here the studies reveal that ACP is actually involved in metabolic homeostasis. The model proposed here is that in response to extended flight, ACP levels are increased in order to cause a metabolic switch in flight muscle. The ability of ACP to perform this switch directly needs to be more comprehensively and quantitatively tested.

The analytical methods are thoughtfully done. The peptidomics works is well designed and successfully analyzes hundreds of neuropeptides in an unbiases manner. This data set will be extremely valuable to researchers.

As the authors point out in their discussion, a metabolic switch to β oxidations is a conserved mechanism in numerous animals especially in the heart during stress. Therefore molecular regulators of this process will have broad impact in various animal models.

A wonderful question about adaptation to extend flights in locusts is asked and a mechanism is suggested via the ACP neuropeptide and its direct actions on flight muscle. The analytical methods are very thoughtful and will be broadly impactful.

This reviewer is appreciative of this elegant piece of work, and has the following questions and suggestions to the authors-

1. The reviewer believes that the genetic model used in this study shows the role of ACP in flight muscle metabolic homeostasis. Therefore as expected the already impaired fatty acid β oxidation can cause the inability to perform extended flights. Does the increased ACP expression actually modulate fatty acid oxidation in response to extended flight as suggested. Please show if there is increased staining of ACP in response to flight.

2. The reviewer also suggests an additional experiment to take a quantitative approach in order to map the causative role of ACP in fatty acid oxidative capacity. This can be done using transgenic animals with inducible expression of ACP and correlation with measuring FAO capacity.

This should also map directly with ability for extended flight strengthening the hypothesis.

*Reviewer #3:*

Neuropeptides are small signalling molecules in the central nervous system that control many physiological and behavioural processes. Hou et al., show that the ACP neuropeptide modulates long-term flight in adult locusts. They first performed a mass spectrometry analysis aiming at identifying all neuropeptides in the locust brain and in its associated neurohaemal organs, the corpora cardiaca-corpora allata. Next, they show that the expression of four neuropeptide-encoding genes changes after 1 hour of sustained flight. Knockdown by RNA interference revealed that down-regulation of two of these genes, respectively encoding for ACP and AKH-2, affect flight time and flight distance. The role of AKH-2 in flight regulation has already been demonstrated, but a possible role of ACP has so far remained elusive. Therefore, the authors made use of CRISPR/Cas9 gene editing to provide compelling evidence that ACP knockouts fail in normal long-term flight activity when compared to wild type controls.

Based on homology searches, the authors then predicted the ACP receptor and found by means of transcriptome analysis that this receptor is highly expressed in the fat body and in-flight muscles of locusts. Comparative transcriptome analysis between wild type and ACP knockout locusts revealed a differential expression of 520 genes in flight muscle and 318 genes in the fat body. The highly expressed FABP gene in flight muscle was downregulated by more than 80% in ACP knockouts, as confirmed by qPCR and Western blotting. RNAi knockdown of the fatty acid binding protein, FABP, suppressed the expression levels of multiple β-oxidation-related genes.

In addition, a comparative metabolite analysis showed that 204 metabolites were downregulated in flight muscles of ACP knockouts, including acetyl-CoA and NADH, two end metabolites of β-oxidation.

RNAi knockdown of FABP resulted in a similar phenotype with decreased flight activity as well as decreased levels of acetyl-CoA and NADH.

Injecting synthetic ACP in ACP knockouts could (partially) rescue expression levels of eight β-oxidation genes as well as metabolite contents and flight activity, but not in FABP knockdown animals, suggesting that FABP may act downstream of ACP.

These data convincingly show that the neuropeptide ACP regulates long-term flight activity in locusts and that the fatty acid transport protein FABP may act downstream of ACP.

A weakness of the study is that it has not been experimentally demonstrated that ACP acts through its predicted ACP receptor in flight muscles. An alternative explanation – namely that ACP induces the release of a flight-controlling hormone from the CC or possibly from the CA into the haemolymph – is still possible.

1) Hou et al., analysed the brain and CC-CA neuropeptidome. For this purpose, they prepared samples of brains and CC-CA. To be able to understand the tissue distribution of the neuropeptides, it is important to know what is meant by 'brain' samples and by CC-CA samples. Does the brain also contain the optic lobes, frontal ganglion, suboesophageal ganglion, the NCC or only the proto-, deutero-, and tritocerebrum? Do the CC-CA samples only contain the CC and the CA or also the hypocerebral ganglion and other small neuronal structures that are closely attached to the CC-CA? The authors should clarify this and compare their data with the neuropeptidomic tissue distribution analysis as described in Clynen and Schoofs, (2009).

2) line 113: In total, 177 and 341 non-redundant neuropeptides derived from 36 precursors were identified in the brain and CC-CA respectively. Does this refer to number of ion peaks, of which several can be derived from the same mature neuropeptide? Or are all distinct endogenously occurring neuropeptides?

3) Figure 1 supplement 1:

- is each of the 36 peptides a representative for each of the 36 neuropeptide precursors? This should be indicated, because there are more than 36 neuropeptides.

- What is Mys-B; Is it locustamyosuppressin? Then it should be abbreviated as MS.

- Ryamine should be RYamide.

- Hou et al., 2015 identified 4 AKH precursor genes. Why is AKH-4 not identified in this analysis?

- Sulfakinin is more abundant in CC-CA compared to brain, which contrast the neuropeptidedome analysis of Clynen and Schoofs, (2009). Explain.

- Allatostatin B should be named as MIP, as this is the now commonly accepted name for the precursor genes of this bilaterian peptide family, of which the first member was identified in Locusta as Lom-MIP.

- Procolin should be Proctolin

- Explain why a number of Locusta neuropeptides, known to be present in the brain and/or CC-CA, were not identified in this neuropeptidome analysis, including (the precursors of) corazonin, AST-C, inotocin.

4) Source data supplementary Table 1.

This table shows the peptide ions corresponding to the neuropeptide precursor genes.

Remarks: Please consult the Locusta neuropeptidomics papers of Clynen and Schoofs and name the peptide precursors according to the originally given neuropeptide names.

– Accession number 12049.1 has 2 names: DH/PBAN and tryptopyrokinin. This is, however, the precursor of locustapyrokinin-1, which was identified by neuropeptidome analysis by Clynen and Schoofs and before. It is confusing to rename the genes encoding pyrokinin peptides. AYC12049 definitely is a locustapyrokinin precursor, and should be named according to the original discovery of its encoding neuropeptides, i.e. PK, and not renamed into DH/PBAN. PK-encoding genes have expanded in the evolutionary lineage leading to locusts, when compared to other insects. Naming them as PK1, PK 2, etc. reflects better this expansion (instead of giving them different names).

– AYC12051.1 seems to be the precursor of the periviscerokinins and should be named PVK or CAPA.

– AKN21242.1 is the precursor for LOM-MIP or locustamyoinhibiting peptide and should not be renamed into AST-B.

– Procolin should be proctolin in the source table of neuropeptides.

– SNPF should be sNPF.

5) Line 191 and following: The authors did not experimentally demonstrate that the ACP neuropeptide (or HrTH) acts through the predicted ACP receptor. The authors can show this in vitro by expressing the receptor in a cellular expression system and challenge the receptor with synthetic ACP.

In addition, to substantiate the mechanism of action of ACP in vivo, the authors should compare the phenotype of the RNAi-knockdown of the ACP receptor (or Crispr-Cas9 knockout) with the flight phenotypes of WT and FABP knockdowns.

6) line 308: The ACP neuropeptide is abundant in the CC (not in the CA) as shown by Clynen and Schoofs (not in this manuscript). The expression of the gene is confined to cells in the brain as shown in this manuscript. Line 308 is about the neuropeptide. Transcription responses refer to the gene expression, which is brain-specific according to Figure 2. Rephrase.

7) line 322: "Our results show that the ACP precursor gene is specifically produced in the brain" This phrasing is incorrect: all cells/tissues contain the same genome. No single gene is specifically produced in any tissue". Should be: specifically expressed.

8) line 329: it is not quite similar. AKH is synthesized in the CC, more particular by cells in the CC's glandular lobes. JH is synthesized in the CA. ACP is synthesised in the brain and transported via the NCC to the storage lobe of the CC, where it is stored (until release) or where it may influence the release of other substances.

6) line 317: The ACP neuropeptide or HrTH was isolated from the storage lobes of the corpora cardiaca. These storage lobes contain peptide-storing nerve endings of the NCC coming from the brain.

7) line 332: rephrase: where its receptor is expressed. (A circulating neurohormone will act on all its receptor molecules on the surface of cells (and not only on tissue displaying a high receptor expression), while circulating through the body).

8) line 365: "Based on transcriptome profile analysis, the flight muscle appears to be the main tissue targeted by ACP, although the ACP receptor also has a relatively high expression level in the fat body".

This is speculation. ACP will also target its receptor in the fat body, where it will elicit a distinct functional response.

Targets of neuropeptides cannot be (exclusively) identified in the context of altered gene expression upon neuropeptide receptor activation. In fact, most neuropeptides act on GPCRs and the responses are usually very rapid (within seconds), while altered gene expression is a process that takes much more time.

Immediately downstream of the neuropeptide-receptor interaction, fast processes are more likely to be the immediate result of this interaction. These fast responses include increased/decreased secretion, change of membrane permeability, opening/closing of ion channels etc. The presence of the ACP receptor in the fat body suggests that – like the flight muscles – also the fat body is a target of ACP, if the predicted ACP receptor is the in vivo cognate receptor of ACP.

The manuscript only provides very weak evidence (the presence of a predicted ACP receptor) that ACP directly acts on flight muscles.

I suggest to delete lines 365 to 380: many interpretations, wrong phrasing. Does not contribute much to the story.

9) line 442: ACP has not been identified in this manuscript. The peptide was identified by Siegert long before and the ACP precursor was already identified in the study of Hou et al., 2015.

Quantitative neuropeptidome analysis:

10) line 476: provide the source of the protein database. Where can it be consulted?

11) why was the enzym trypsin set as a parameter in a peptidomics analysis? This is done for tryptic peptide analysis in a classical proteome analysis, but it is not the optimal way of analysing the neuropeptidome. This parameter does not yield the theoretical spectra of mature neuropeptides. Instead, it yields theoretical spectra of theoretic tryptic peptides, which aid in the identification of the corresponding protein. Spectra of tryptic peptides do not reveal the endogenously occurring neuropeptides, which are endogenously processed by other specific processing enzymes.

12) line 482: what is meant by this? At least two spectra in one sample?

13) What is the mass tolerance for fragmented peptides? Modifications should also include C-terminal amidation, N-terminal pGlu.

14) Figure 3 supplement 2: Explain abbreviations. The expression of ACPR is only tissue-specific for the organs/tissues tested. (The gut, Malpighian tubules, trachea, hypodermis, etc were not analysed). I suggest to rephrase in manuscript text as: Compared to the other organs tested, ACPR is highly expressed in flight muscles and fat body.

15) line 567: include reference or source of MEGA software.

16) Bogerd et al., (1995) showed that flight activity increased steady-state levels of the AKH I and II mRNAs (approximately 2.0 times each) and the AKH III mRNA (approximately 4.2 times) in the corpora cardiaca. Explain why an altered expression of AKH-1 and AKH-III precursor genes is not seen in the present manuscript.

17) Many (supplemental) Figures contain abbreviations that are not explained in the Figure legends.

[Editors' note: further revisions were suggested prior to acceptance, as described below.]

Thank you for submitting your article "Neuropeptide ACP facilitates lipid oxidation and utilization during long-term flight in locusts" for consideration by *eLife*. Your article has been reviewed by 3 peer reviewers, one of whom is a member of our Board of Reviewing Editors, and the evaluation has been overseen by K VijayRaghavan as the Senior Editor. The following individual involved in review of your submission has agreed to reveal their identity: Aravind Ramanathan (Reviewer #2).

Essential Revisions:

Thank you for clarifying the peptidomic analysis in your revised version.

1. However it is essential that your analysis makes clear the source of brain region used for the peptidomic analysis. Therefore "CC-CA" should be replaced by "retrocerebral complex" in the entire manuscript.

2. Also, rewrite lines 370-375, by stating that in the present manuscript the whole retrocerebral complex was used for peptidomic analysis, in contrast to the study of Clynen who analysed the organs of the retrocerebral complex separately.

Delete "a little bit different" (it is not a little bit).

3. An important methodology consideration, that is key to interpreting your results has been raised. Please address this in your revision.

– The authors have adapted the methods section on mass spectrometry. It is much clearer now, but one major issue remains. It regards the sentence "Neuropeptide identification were used only if at least two spectra were identified in one sample".

This is a criterion that is used for proteomics analyses; it is not at all suited for neuropeptidomics studies. The reason is that neuropeptides are produced by cleavage from larger preproteins, whereby the mature cleaved peptide is kept and the remainder of the protein is degraded. So, if the preprotein contains only one mature neuropeptide sequence, this neuropeptide cannot be identified by the restricted rule of "two spectra in one sample". The authors refer in their comments to the publication of Han et al. in J. Protein Research. But in the discussion of this article on page 4389, Han et al. clearly state that the 19 neuropeptides that they were unable to identify might be caused by the stricter FDR threshold they used (two spectra in one sample). That is the reason that they missed 19 well known bee neuropeptides in their study.

Regarding the present revised manuscript one may wonder in fact how the authors were even able to identify the ACP neuropeptide by using this restrictive threshold. Because the MS-spectra were not provided in the MS (I could not find them), I cannot check what other peptide from the same ACP-precursor they identified. This would be strange, because the remainder of the protein is normally broken down upon cleavage of the ACP peptide out of it.

So the question is: did the authors use this restrictive threshold? (which might also explain why they were unable to identify some well known locust peptides residing as single peptides in their respective precursors).

If yes, then this reviewer would like to know what other peptide they identified from the ACP precursor, as this would contradict with textbook knowledge about neuropeptide processing. In this aspect, the term "lots of new peptide fragments" is unclear. What is meant by "peptide fragments"? Are these fragments of the same mature peptide or fragments of the remainder of the precursor protein, which should in principle have been degraded.

Or did they not use this restrictive threshold?

[Editors' note: further revisions were suggested prior to acceptance, as described below.]

Thank you for resubmitting your work entitled "Neuropeptide ACP facilitates lipid oxidation and utilization during long-term flight in locusts" for further consideration by *eLife*. Your revised article has been evaluated by K VijayRaghavan (Senior Editor) and a Reviewing Editor.

The manuscript has been improved but there are some remaining issues that need to be addressed, as outlined below:

Reviewer #1:

Once Reviewer #3 approves the responses of the authors, I am fine with it. Reviewer #3 is the expert in regard to these specific points

Reviewer #3:

I still have a major concern regarding the neuropeptidome analysis, which I explain in detail below.

The authors state in their rebuttal : " Based on the reviewer's comments, we realize that our description of neuropeptide identifications is inaccuracy in the sentence "Neuropeptide identifications were used only if at least two spectra were identified in one sample". To avoid misunderstanding, we have revised the description as: The criterion of peptide fragment identifications was used only if at least two spectra were formed in one sample [...] Generally, a single peptide fragment could be detected for several times by the MS. A spectrum will be formed each time when the peptide fragment is detected by MS. The criterion of peptide fragment identifications was used only if at least two spectra were formed in one sample"

In the text, the authors wrote: ""The threshold used in the current study (two spectra in one sample) could significantly improve the reliability of peptides identified in the MS. "

Reply:

1) This statement is wrong. One MSMS spectrum of good quality is sufficient for the identification of a neuropeptide. An MSMS fragmentation spectrum of good quality is a spectrum where the series of y-ions and b-ions are sufficiently complete to cover the whole amino acid sequence of the peptide (which is the case for the spectra nrs 1 and 4 provided by the authors, but not for spectrum 5).

You can never state that at least two MSMS spectra of the same neuropeptide are needed. The argument of the authors "more peptide spectra = more reliability" is also not in line with the way data are collected on the mass spec.

In data-dependent acquisition, which is how the mass spec collects data, you have to take into account the dynamic exclusion parameter, which ensures that an ion with a particular m/z value will not be selected for fragmentation for a number of seconds to avoid that the instrument detects and measures the same ion over and over again during an elution peak. The normal situation with a dynamic exclusion parameter setting at 15s, is that the mass spectrometer detects a single peptide ion, because peptides typically elute over 30s. The detection of many identical ions of the same peptide is the exception rather than the normal situation and should not be set as a rule of confidence. The dynamic exclusion parameter cannot be found in the methods section.

So, in the normal situation, only one precursor ion is selected for MSMS fragmentation. Only when the elution peak of that peptide ion is broad, the peptide ion can be selected multiple times over the period of the elution peak. That is why the dynamic range parameter is important. This parameter avoids that the same ion is selected for fragmentation over and over again at the expense of other – less abundant – peptide ions that elute in the very same region.

The "more than one spectrum in one sample" argument of the authors is thus a reflection of the elution and abundance of a particular peptide ion. It is not at all a criterion for reliability or confidence regarding the MSMS peptide identification. One spectrum of good quality is as reliable as two or three good spectra of the same ion that just elutes over a longer period of time.

The omission of neuropeptides represented by a single MSMS spectrum of good quality is thus a mistake.

A second mistake is that all the measured neuropeptide degradation products are counted as bona fide neuropeptides (see below under nr 2).

The four ACP spectra provided in the rebuttal are all spectra of the same ACP sequence. The excel file also contains degradation products of this peptide. However, there is no reason to assume that the measurement of multiple spectra in one sample or the measurement of peptide degradation products provide a higher reliability. The measurement of degradation products can be due to a technical issue (sample preparation) or a biological issue (in vivo degradation). It does not tell you anything about the reliability of the measurement of the intact neuropeptide.

2) Results section

The authors state: "In total, 177 and 341 nonredundant neuropeptides derived from 36 precursors were identified in Br and RC, respectively (Supplementary File 1).

Reply:

Now that the authors have explained how they have interpreted the MSMS data, this sentence is completely wrong and will lead to confusion. The authors have not identified that high number of neuropeptides, because this number reflects multiple MSMS spectra resulting from the same neuropeptide (and its degradation products). A neuropeptide is defined as the amino acid sequence located between the dibasic cleavage sites within the neuropeptide precursor. The shorter versions presented in the excel file are not novel neuropeptides but degradation products. It is unclear what the origin of these degradation products is. They may be artifacts resulting from the sample preparation, or they may represent in vivo degradation fragments resulting from the inactivation of the neuropeptide.

The authors should rearrange their excel file by clustering the degradation products of the same neuropeptide within a neuropeptide precursor. Here an example for a pyrokinin (PK1)-precursor derived neuropeptide.

GAVPAAQFSPRLamide

GAVPAAQFSPRL

AVPAAQFSPRLamide

AVPAAQFSPR

VPAAQFSPRL

VPAAQFSPR

PAAQFSPRLamide

AQFSPRLamide

3) The authors also state: " In addition, only several of known neuropeptides, not too many, were not detected under the current threshold in our study. The absence of these neuropeptides may be attributed to their low abundance in tested samples, relative short half-life period, and unsuitable chromatographic conditions. We think that the criterion used in the current study should be suitable."

Reply:

The identification of the neuropeptides is also dependent on the used database for spectrum scanning, as the authors did not rely on de novo identification. Thus, the identification can only be as good as the database against which the spectra are scanned. If the used locust database does not contain the (protein precursor) sequences, or the derived theoretical spectra, particular experimental spectra will not lead to identification, even if they are abundantly present;

The second reason is due (i) to their data acquisition setting (parameter not mentioned in the methods) and to (ii) the "two spectra in one sample" rule that the authors wrongly used, a rule that merely reflects long elution times of very highly abundant peptides (see above). This rule is wrong and excludes less abundant neuropeptides with clear but shorter chromatographic elution peaks.

4) In the excel files, CC-CA has still not been replaced by retrocerebral complex.

---

## [Author Response]

Essential Revisions:1) While a potential ACP receptor is presented on the basis of informatics and expression analysis, there is no functional evidence of a role for this protein in mediating the actions of ACP. The authors can show this in vitro by expressing the receptor in a cellular expression system and challenge the receptor with synthetic ACP.AND/ORto substantiate the mechanism of action of ACP in vivo, the authors should compare the phenotype of the RNAi-knockdown of the ACP receptor ( or Crispr-Cas9 knockout) with the flight phenotypes of WT and FABP knockdowns.

We accept the reviewer’s comments. In fact, we have successfully constructed the homozygous ACPR mutant locust line using the CRISPR/Cas 9 system. As expected, the ACPR mutant (ACPR^13/13^) indeed displayed significantly reduced long-term flight activity, similar to the flight phenotype caused by *ACP* knockout and *FABP* knockdown. The experimental evidence about molecular and functional identifications of ACPR has been provided in the revised Figure 3.

2) A key strategy in this study is to compare phenotypes, transcriptomes and metabolomes between wild type and ACP-/- animals. A body size defect is reported for ACP-/- locusts that is also seen with the RNAi. This is an important point as in insects adult body size is determined during development and will also imply effects of metabolism. In general, studying rapidly adaptive processes like metabolism (that change based on seasons and nutritional availability) using knockout lines can lead to alternative metabolic adaptations that will be difficult to map. This is a concern in an otherwise elegant study.

We understand the reviewer’s concern. However, the body size of locusts actually does not change after adult eclosion thanks to its hard exoskeleton. Therefore, the effects of ACP on body size could not be assessed through RNAi of the gene after adult eclosion. So, we have clarified this point by providing additional discussion in the revised manuscript. Details were shown as: “Through phenotype examination, we also observed a larger body size in ACP mutants. Usually, the body size of locusts is stable after adult eclosion thanks to its hard exoskeleton. Therefore, the effects of ACP on body size could not be assessed through RNAi of the gene after adult eclosion in the current study. It has been suggested that the growth state for an organism can be negatively affected by other physiological traits, such as locomotion, reproduction, or life span (Lee et al., 2010). Therefore, the increased body size of ACP mutants may be attributed to the continuous metabolism changes associated with trade-off effects between flight activity and body growth. Similarly, the loss of function of the AKH peptide results in adult-onset obesity in *Drosophila* (Galikova et al., 2015). These findings may reflect a common role played by ACP and AKH in governing the energy balance of insects. It will be an interesting work to explore the molecular and metabolic basis for body size determination on the basis of established ACP mutant locust line” (lines 503-515).

The authors should strengthen the conclusion that enhanced ACP activity in adult locusts is responsible for the reported transcriptome and metabolome changes leading long flight modulation. Can they provide data that:a) Please show if there is altered staining of ACP in the brain during long flight conditions?

We appreciated the reviewer’s suggestion. Actually, we have tried to determine the intensity of ACP staining in response to 60 min-sustained flight by performing IHC experiments before. However, it seems that the IHC method was unsuited to do quantitative analysis of ACP staining. Several reasons might be taken into considerations: (1) There are many ACP-positive neurons (>30 neurons in the pars intercerebralis and > 8 neuron in each lateral forebrain), and some of them showed overlapped staining, which could lead to inaccurate intensity analysis; (2) Although the cellular localization of ACP peptide is in similar brain regions, numbers of ACP-positive staining neurons varied among different individual brains thanks to the complex stereoscopic neuronal structures (see Author response image 1, CK indicates control locust brains, FL indicates brains from locusts after 1 h-sustained flight). In fact, the evidence from the significant increase of *ACP* transcription after 1h-sustained flight (in Figure 2B) indicates the sensitive response of ACP to flight activity. We have discussed this point in the Discussion: “Our results show that the *ACP* precursor gene in the brain displays strong transcription responses to prolonged flight. The regulatory roles played by ACP in locust flight are clearly supported by tethered flight experiments after knockdown and knockout of its precursor gene, as well as reduced extended flight ability of ACPR mutants” (lines 390-394).

b) Does the increased ACP expression actually modulate fatty acid oxidation in response to extended flight as suggested. Can injection of ACP peptide alter FAO capacity and map to extended flight capability?

Thanks for the reviewer’s suggestions. According to this suggestion, we added a new experiment to examine the flight performance and β-oxidation of WT locusts through the injection with ACP peptide. The results showed that the injection of ACP peptide could significantly promote locust flight activity including both flight duration and flight distance (new Figure 1—figure supplement 7). Moreover, the expression levels of lipid oxidation-related genes and the abundance of β-oxidation products in the flight muscle were also significantly enhanced upon ACP peptide injection (new Figure 4—figure supplement 3A and Figure 5—figure supplement 4A). Therefore, these new data further support that increased ACP level can promote fatty acid oxidation and facilitate sustained flight in turn.

c) If the changes in transcriptome and metabolome seen in ACP-/- are a direct reflection of the function of this peptide is modulating metabolism in long-flight, then are the changes in these transcripts and metabolites also seen in wild type locust tissue when comparing resting with tissue after 1 hr flight ?

We understand the reviewer’s concern. In the revised manuscript, we performed additional experiment to examine the expression levels of genes associated with fatty acid transport and oxidation and the contents of end metabolites of β-oxidation (acetyl-CoA and NADH) in the flight muscle after 1 h-sustained flight as well as ACP peptide injection. We found both 1 h- sustained flight and ACP peptide injection could significantly enhance the expressions levels of gene related to lipid transport and oxidation in the flight muscle of WT locusts (Figure 4—figure supplement 3). Moreover, injection of ACP peptide could strongly increase the relative amounts of muscle acetyl-CoA and NADH those reflecting fatty acid oxidation activity (Figure 5—figure supplement 4A), further supporting the regulatory role of ACP peptide in lipid metabolism. However, upon 60 min-tethered fight, the relative content of acetyl-CoA did not change but the NADH level was decreased (Figure 5—figure supplement 4B). The decreased β-oxidation products may reflect rapid energy utilization in subsequent mitochondrial metabolism during sustained flight. The consistent expression changes in FAO-related genes upon ACP peptide injection and sustained flight could further support the tight regulation of ACP on flight-related lipid oxidation. We have provided additional discussion on this point in the revised manuscript (see lines 433-440).

3) Clarity on methodologyPlease provide clarity on some methodology issues to make the work more informative and accessible. The following points to be addressed:i) Hou et al. analysed the brain and CC-CA neuropeptidome. For this purpose, they prepared samples of brains and CC-CA. To be able to understand the tissue distribution of the neuropeptides, it is important to know what is meant by 'brain' samples and by CC-CA samples. Does the brain also contain the optic lobes, frontal ganglion, suboesophageal ganglion, the NCC or only the proto-, deutero-, and tritocerebrum? Do the CC-CA samples only contain the CC and the CA or also the hypocerebral ganglion and other small neuronal structures that are closely attached to the CC-CA?

We appreciated the reviewer’s helpful suggestions. In the current study, brain samples containing only the protocerebrum, deuterocerebrum, and tritocerebrum and the corpora cardiaca-corpora allata (CC-CA) complex including attached small neuronal structures and hypocerebral ganglion were used for neuropeptidome analysis. We have provided detailed information in the method part (lines 532-534).

ii) Although the neuropeptidomics / mass spectrometry data do not directly influence the conclusions of the manuscript, it is crucial that the authors clarify the trypsin setting in their MS analysis. Because this may be a major mistake. It is hard to believe that the neuropeptidomics data were generated by this method. Explanation: Trypsin generates theoretical peptide spectra that end in R or K. So it is impossible that these theoretical tryptic peptide spectra were used for the identification of naturally occurring neuropeptide spectra (as these do not by definition end in R or K).

We really appreciate the reviewer’s comments and apologize for this mistake. We have corrected it in describing the neuropeptidome method. Detailed methods for neuropeptidome were shown as: “parent ion mass tolerance is 15 ppm, and fragment ion mass tolerance is 0.05 Da; enzyme specificity, none. The following modifications were applied: C-terminal amidation (A, −0.98) and pyroglutamination from Q (P, −17.03), maximum missed cleavages per peptide: 2, and maximum allowed variable PTM per peptide: 2. A fusion target and decoy approach was used to for the estimation of the false discovery rate (FDR) and controlled at ≤1.0% at the peptide level. Neuropeptide identifications were used only if at least two spectra were identified in one sample. Peptides were further validated by comparison with predicted neuropeptide precursors and neuropeptidome analysis in the locust (Clynen and Schoofs, 2009, Hou et al., 2015)” (lines 558-567).

iii) Please revise the annotation of the peptidomic data in line with the conventions or existing literature for the field so that this work can be used in conjunction with existing literature. The authors should compare their data with the existing neuropeptidomic tissue distribution literature, for e.g in Clynen and Schoofs (2009).(detailed comments on this point provided in the individual reviews below).

Thanks, we have carefully compared our peptidome data with that reported by Clynen and Schoofs, (2009). The annotation of neuropeptides has been revised in corresponding figures and tables according to the reviewer’s suggestion.

iv) It is not clear if the transcriptome analysis was performed on resting muscle tissue or on tissue collected after 1 hr flight. Same applies to metabolomics analysis.

In the current study, the transcriptome analysis was performed on resting muscle tissue. We have added detailed information in the method part: “The flight muscle and fat body tissues were dissected from WT and ACP^-/-^ female locusts under resting state at PAE 7 days” (lines 671-672).

v) In all figure notations such as a, ab, b, bc , abc have been used. It is apparently related to statistical analysis but this is very confusing and not well explained. The statistical analysis in all figure should be explained much more clearly.

Multiple comparison between groups were analyzed using one-way ANOVA method. Columns labeled with different letters indicate that there is a significant difference between the two groups, columns contain same letters indicate no significance observed between the two groups. We have explained the meaning of letters labeled on different columns in the corresponding figure legends (please see lines 1051-1053, 1098-1101).

Reviewer #1:To identify neuropeptides that might regulate muscle metabolism during sustained flight, the authors carried out mass spectrometry to identify peptides that are expressed in the brain and neuroendocrine tissues using high-resolution mass spectrometry. From the peptides found in these tissues, they shortlisted those (ACP, AKH2 and NPF1) whose levels were elevated during long-term flight; of these, when depleted by RNAi, ACP and AKH2. A role for AKH2 in controlling metabolism and flight has been previously described. Here the authors study the role of ACP in this process.To study the role of ACP the authors deleted the gene using CRISPR/Cas9 gene editing and demonstrate loss of the peptide in edited animals by immunohistochemistry. Knockout animals were larger than controls and showed shorter flight times and distances travelled. This suggests a role for ACP in regulating long term flight. Using bioinformatics analysis, the authors identify the likely candidate receptor for ACP (ACPR) that could mediate the actions of ACP. It is highly expressed in muscle and fat body. However the functional significance of this receptors in terms of ACP action remains to be established.To understand the mechanism by which ACP supports flight a transcriptome analysis was carried out on flight muscle and fat body tissue comparing wild type and ACP-/-. From a statistical analysis the authors conclude that the likely target is muscle although this remains to be established. Pathway analysis of transcript changes indicate alteration in molecules involved in metabolism. Perhaps an analysis of transcriptional changes in wild type muscle will help to identify transcript changes relevant to adaptation to long term flight. A key molecule identified in this analysis was FABP, a key transporter of fatty acids. The functional importance of this molecule was determined by the RNAi depletion of FABP in muscle which resulted in defects in a tethered flight assay.To determine the impact of these transcriptional changes ACP-/- muscles on metabolism, a metabolome analysis was performed on muscle tissue using LC-MS. A key finding was the downregulation of several molecular species of acylcarnitines with no change in the levels of pyruvate, the end product of glycolysis. Together with the transcriptome analysis, this study concludes that ACP signalling during long flight remodels metabolism to enhance mitochondrial metabolism.The definition of long-term flight as used in the context of this study and the phenomenon being discussed have not been clearly defined. This should be done early on in the manuscript.

We accepted the reviewer’s suggestion. We have provided the definition of long-term flight in the introduction. Details were presented as: “Long-term flight is usually defined as sustained flight for seasonal and long-range migration towards a distinct direction in populations (Stefanescu et al., 2013, Juhasz et al., 2020)” (lines 50-52).

While the manuscript makes a case for the function of ACP, what is the role of AKH2. Clearly not sufficient to account for loss of ACP. What is the phenotype of a depletion of both AKH2 and ACP. AKH2 Can RNAi for AKH2 be done in ACP-/- animals and if so what would be the flight defect ?

Yes, this is an interesting question to explore the interaction of AKH2 and ACP in modulating the flight activity of locusts. In fact, previous documents have demonstrated the role of AKH members in locust flight by lipid mobilization of fat body (Van der Horst, 2003; Bogerd et al., 1995). It is reasonably hypothesized that there would be the coordination between the two peptides in the regulation of long-term flight of locusts. We are willing to test this hypothesis in the future work. In the revised manuscript, we have enriched the discussion about this point. The details were presented as: “Here, we also revealed that knockdown of either *ACP* or *AKH2* induced similarly suppressed effects on locust flight performance. Further work is warranted to investigate the potential interaction among these neuroendocrine factors in energy regulation associated with flight activity” (lines 499-502).

A body size defect is reported for ACP-/-. Is this also seen with the RNAi. This is an important point as in insects adult body size is determined during development and will also imply effects of metabolism. Please clarify this point.

Thanks for the reviewer’s comments. Actually, the body size of locusts is stable after adult eclosion thanks to its hard exoskeleton. Because our RNAi treatments were conducted in adult locusts, the effects of ACP on body size could not be seen in the current study. We have clarified this point by providing additional discussion in the revised manuscript (lines 503-515).

What is the effect of RNAi depletion, in muscle, of ACPR on flight duration and length. Can it phenocopy the ACP depletion or knockdown ? At the moment this data is of limited value.

We accepted the reviewer’s concern. We have recently successfully constructed the homozygous ACPR mutant locust line using the CRISPR/Cas 9 system. The ACPR mutant (ACPR^13/13^) indeed induced similar phenotypes as ACP mutants, and displayed significantly reduced flight time and flight distance. The experimental evidence related to molecular and functional identification of ACPR were presented in the revised Figure 3.

It is not clear if the transcriptome analysis was performed on resting muscle tissue or on tissue collected after 1 hr flight. Same applies to metabolomics analysis.

The muscle tissues used in both transcriptome and metabolomics analysis were collected from WT and ACP^-/-^ locusts under resting state. We have clarified this information in the method part in the revised manuscript (lines 671-672, 690-691).

Also what are the transcriptome changes in wild type muscle tissue comparing resting and 1 hr flight. This information will help identify the changes relevant to long term flight in the ACP-/- and separate out transcriptional changes that are developmental compared to wild type.

We thanked the reviewer’s valuable suggestion. In the current study, we examined the transcriptome changes in the flight muscle of ACP^-/-^ locusts compared to WT locusts. In the revised manuscript, to further clarify the reviewer’s concern, we systematically examined the expression levels of genes associated with fatty acid transport and oxidation in the flight muscle after 1 h-sustained flight using qPCR methods. We found the expressions levels of genes related to lipid oxidation in the flight muscle were significantly up-regulated upon sustained flight. These results suggest that transcriptional changes in metabolism related genes are tightly correlated with flight activity. Relative experimental data were show in (Figure 4—figure supplement 3B).

In all figure notations such as a, ab, b, bc , abc have been used. It is apparently related to statistical analysis but this is very confusing. The statistical analysis in all figure should be explained much more clearly.

Thanks for the reviewer’s suggestion. One-way ANOVA method was used in our multiple comparisons between groups. Columns labeled with different letters indicate that there is a significant difference between the two groups, columns contain same letters indicate no significance observed between the two groups. To more clarify this concern, we have explained the meaning of letters labeled on different columns in the corresponding figure legends.

Reviewer #2:– The authors aim to understand how locusts modulate skeletal muscle metabolism to support long flights that is a well known aspect of the animal's natural behavior. The authors hypothesize that a neuropeptide directly signals to flight muscles and promote a metabolic switch (fatty acid β oxidation) that can support long flights. The authors aimed to identify one or more neuropeptides that can perform this metabolic switch.– In order to identify the neuropeptide the authors used a combination of mass spectrometry based peptidomics, mRNA profiling and RNAi mediated knockdown studies and immunofluorescence staining. This part of the study identified 2 neuropeptides ACP and AKH2 that were upregulated in response to sustained flight. Since ACP was less studied in the context of this study, the authors proceeded with this peptide for functional studies.– Using CRISPR knockout lines, the authors showed the functional dependency of this peptide on sustained flight.– They identified the role of this peptide in promoting β oxidation of fatty acids (using metabolomics and mRNA profiling) as judged by impaired β oxidation in ACP knockout animals. This effect was recued by peptide treatment.– The authors next showed a specific FABP protein as a mediator of the effects of ACP in promoting fatty acid transport and β oxidation in response to this neuropeptide.The systems analysis of metabolism is a major strength of this paper.In general, studying rapidly adaptive processes like metabolism (that change based on seasons and nutritional availability) using knockout lines can lead to alternative metabolic adaptations that will be difficult to map. This is a concern in an otherwise elegant study. Though not highlighted here the studies reveal that ACP is actually involved in metabolic homeostasis. The model proposed here is that in response to extended flight, ACP levels are increased in order to cause a metabolic switch in flight muscle. The ability of ACP to perform this switch directly needs to be more comprehensively and quantitatively tested.

We appreciate the reviewer’s comments. In the revision, we have provided addition experimental data to support the regulatory role in lipid metabolism related to prolonged flight. We examined the flight performance and β-oxidation activity of WT locusts after injection with ACP peptide (new Figure 1—figure supplement 7, new Figure 4—figure supplement 3A, and Figure 5—figure supplement 4A). The enhanced prolonged flight activity, increased expressions levels of gene related to lipid utilization, as well as the elevated abundance of β-oxidation products in the flight muscle upon ACP peptide injection could further support the essential role of ACP in muscle lipid utilization during flight.

The analytical methods are thoughtfully done. The peptidomics works is well designed and successfully analyzes hundreds of neuropeptides in an unbiases manner. This data set will be extremely valuable to researchers.As the authors point out in their discussion, a metabolic switch to β oxidations is a conserved mechanism in numerous animals especially in the heart during stress. Therefore molecular regulators of this process will have broad impact in various animal models.A wonderful question about adaptation to extend flights in locusts is asked and a mechanism is suggested via the ACP neuropeptide and its direct actions on flight muscle. The analytical methods are very thoughtful and will be broadly impactful.This reviewer is appreciative of this elegant piece of work, and has the following questions and suggestions to the authors-1. The reviewer believes that the genetic model used in this study shows the role of ACP in flight muscle metabolic homeostasis. Therefore as expected the already impaired fatty acid β oxidation can cause the inability to perform extended flights. Does the increased ACP expression actually modulate fatty acid oxidation in response to extended flight as suggested. Please show if there is increased staining of ACP in response to flight.

We appreciated the reviewer’s concern. Actually, we have tried to determine the intensity of ACP staining in response to 1 h-sustained flight by performing IHC experiments before. However, it seems that the IHC method was unsuited to do quantitative analysis of ACP peptide. Several reasons might be taken into considerations: (1) There are many ACP-positive neurons (>30 neurons in the pars intercerebralis and > 8 neuron in each lateral forbrain), and some of them showed overlapped staining, which could lead to inaccurate intensity analysis; (2) Although the cellular localization of ACP peptide is in similar brain regions, numbers of ACP-positive staining neurons varied among different individual brains thanks to the complex stereoscopic neuronal structures (see Author response image 1). In fact, the evidence from the significant increase of ACP transcription after 1h-sustained flight (in Figure 2B) indicates the sensitive response of ACP to flight activity. We have discussed this point in the Discussion “Our results show that the *ACP* precursor gene in the brain displays strong transcription responses to prolonged flight. The regulatory roles played by ACP in locust flight are clearly supported by tethered flight experiments after knockdown and knockout of its precursor gene, as well as reduced extended flight ability of ACPR mutants.”

2. The reviewer also suggests an additional experiment to take a quantitative approach in order to map the causative role of ACP in fatty acid oxidative capacity. This can be done using transgenic animals with inducible expression of ACP and correlation with measuring FAO capacity.This should also map directly with ability for extended flight strengthening the hypothesis.

We appreciated the reviewer’s helpful suggestion. So far, it is not available to overexpress target genes using the transgenic technology in locusts. Instead, we have provided another experimental data to map the causative role of ACP in fatty acid oxidative capacity. We have tested the flight performance and fatty acid oxidative capacity in the fight muscle by artificially injecting synthetic ACP peptide in the ACP mutants. The results showed that the impaired prolonged flight performance and fatty acid oxidation (at both gene expression level and metabolite level) in ACP mutants could be significantly recovered by ACP peptide injection. Furthermore, we have provided addition experimental evidence showing that the artificial injection of ACP peptide in WT locusts could also significantly enhance flight performance (new Figure 1—figure supplement 7) as well as fatty acid oxidation capacity (new Figure 4—figure supplement 3A and Figure 5—figure supplement 4A).

Reviewer #3:Neuropeptides are small signalling molecules in the central nervous system that control many physiological and behavioural processes. Hou et al., show that the ACP neuropeptide modulates long-term flight in adult locusts. They first performed a mass spectrometry analysis aiming at identifying all neuropeptides in the locust brain and in its associated neurohaemal organs, the corpora cardiaca-corpora allata. Next, they show that the expression of four neuropeptide-encoding genes changes after 1 hour of sustained flight. Knockdown by RNA interference revealed that down-regulation of two of these genes, respectively encoding for ACP and AKH-2, affect flight time and flight distance. The role of AKH-2 in flight regulation has already been demonstrated, but a possible role of ACP has so far remained elusive. Therefore, the authors made use of CRISPR/Cas9 gene editing to provide compelling evidence that ACP knockouts fail in normal long-term flight activity when compared to wild type controls.Based on homology searches, the authors then predicted the ACP receptor and found by means of transcriptome analysis that this receptor is highly expressed in the fat body and in-flight muscles of locusts. Comparative transcriptome analysis between wild type and ACP knockout locusts revealed a differential expression of 520 genes in flight muscle and 318 genes in the fat body. The highly expressed FABP gene in flight muscle was downregulated by more than 80% in ACP knockouts, as confirmed by qPCR and Western blotting. RNAi knockdown of the fatty acid binding protein, FABP, suppressed the expression levels of multiple β-oxidation-related genes.In addition, a comparative metabolite analysis showed that 204 metabolites were downregulated in flight muscles of ACP knockouts, including acetyl-CoA and NADH, two end metabolites of β-oxidation.RNAi knockdown of FABP resulted in a similar phenotype with decreased flight activity as well as decreased levels of acetyl-CoA and NADH.Injecting synthetic ACP in ACP knockouts could (partially) rescue expression levels of eight β-oxidation genes as well as metabolite contents and flight activity, but not in FABP knockdown animals, suggesting that FABP may act downstream of ACP.These data convincingly show that the neuropeptide ACP regulates long-term flight activity in locusts and that the fatty acid transport protein FABP may act downstream of ACP.A weakness of the study is that it has not been experimentally demonstrated that ACP acts through its predicted ACP receptor in flight muscles. An alternative explanation – namely that ACP induces the release of a flight-controlling hormone from the CC or possibly from the CA into the haemolymph – is still possible.

We accepted the reviewer’s concern. To clarify this concern, we have provided new experimental data and evidence to support the essential role of ACP peptide system using ACPR mutant. We have recently successfully constructed the homozygous ACPR mutant locust line using the CRISPR/Cas 9 system. The ACPR mutant (ACPR^13/13^) indeed displayed significantly reduced long-term flight activity, which phenocopied the flight performance caused by ACP knockout. The experimental evidence related to molecular and functional identification of ACPR was presented in the revised Figure 3.

1) Hou et al., analysed the brain and CC-CA neuropeptidome. For this purpose, they prepared samples of brains and CC-CA. To be able to understand the tissue distribution of the neuropeptides, it is important to know what is meant by 'brain' samples and by CC-CA samples. Does the brain also contain the optic lobes, frontal ganglion, suboesophageal ganglion, the NCC or only the proto-, deutero-, and tritocerebrum? Do the CC-CA samples only contain the CC and the CA or also the hypocerebral ganglion and other small neuronal structures that are closely attached to the CC-CA? The authors should clarify this and compare their data with the neuropeptidomic tissue distribution analysis as described in Clynen and Schoofs, (2009).

We appreciated the reviewer’s helpful suggestions. We have provided detailed information in the method part (lines 532-534). Moreover, we have carefully compared tissue distribution of neuropeptides identified in current study with the previously work described by Clynen and Schoofs, (2009) according to the suggestion. We found that most of the neuropeptides show similar tissue distribution in the two studies, except for sulfakinin and PVK. The discrepancy in neuropeptidome analysis between the two studies may be attributed to different sample collection strategies. In neuropeptidome analysis of our study, the brain tissue contains protocerebrum, deuterocerebrum and tritocerebrum, and the corpora cardiaca-corpora allata (CC-CA) complex attached some of around neuronal tissues, a little bit different with previous study reported by Clynen and Schoofs, in which distinct neuronal regions (protocerebrum, pars intercerebralis, tritocerebrum, CC, CA) were separately collected for neuropeptidome analysis. We have added detailed information in the Discussion (See lines 352-377).

2) line 113: In total, 177 and 341 non-redundant neuropeptides derived from 36 precursors were identified in the brain and CC-CA respectively. Does this refer to number of ion peaks, of which several can be derived from the same mature neuropeptide? Or are all distinct endogenously occurring neuropeptides?

The numbers of non-redundant neuropeptides refer to distinct endogenously occurring neuropeptides identified in the current study.

3) Figure 1 supplement 1:- is each of the 36 peptides a representative for each of the 36 neuropeptide precursors? This should be indicated, because there are more than 36 neuropeptides.

In Figure1-supplement 1, each line represents the content of neuropeptides produced from a single precursor. We have revised the figure legend as: “the abundant level of neuropeptides generated from 36 neuropeptide precursors were log transformed for heat map drawing” (lines 1190-1191).

- What is Mys-B; Is it locustamyosuppressin? Then it should be abbreviated as MS.

Thanks, we have revised Mys-B as MS according to the reviewer’s suggestion (see Figure 1—figure supplement 1).

- Ryamine should be RYamide.

Thanks, we have revised Ryamine as RYamide.

- Hou et al., 2015 identified 4 AKH precursor genes. Why is AKH-4 not identified in this analysis?

We previously identified 4 AKH precursor genes based on the locust genome and transcriptome sequences (Hou et al., 2015). However, the predicted AKH4 peptide (QVTFSRDWSP) has not been identified by mass spectrometry analysis so far. One possible explanation is that the peptide abundance is very low and beyond the detection threshold of mass spectrometry. We have added more information in the results (Lines 358-365).

- Sulfakinin is more abundant in CC-CA compared to brain, which contrast the neuropeptidedome analysis of Clynen and Schoofs, (2009). Explain.

We accepted the reviewer’s concern. The discrepancy in the tissue distributions of sulfakinin between the two studies may be resulted from different sample collection strategies. In the current study, we used brains containing protocerebrum, deuterocerebrum and tritocerebrum, as well as corpora cardiaca-corpora allata complex (CC-CA) including attached neuronal structures for neuropeptidome analysis, whereas protocerebrum, pars intercerebralis, tritocerebrum, CC and CA were separately collected for neuropeptidome analysis in the other study. Moreover, the developmental stages of experimental insects used are different in the two studies: mature female adults for our study, whereas immature adults for Clynen and Schoofs’s work. We have added more information in the discussion to clarify this concern (lines 366-377).

- Allatostatin B should be named as MIP, as this is the now commonly accepted name for the precursor genes of this bilaterian peptide family, of which the first member was identified in Locusta as Lom-MIP.

We have revised Allatostatin B as MIP in the relative figures and tables as the reviewer’s suggestion.

- Procolin should be Proctolin

Thanks, we have corrected the word in relative figures and texts.

- Explain why a number of Locusta neuropeptides, known to be present in the brain and/or CC-CA, were not identified in this neuropeptidome analysis, including (the precursors of) corazonin, AST-C, inotocin.

We accepted the reviewer’s query. Actually, the detection of small peptides by mass spectrometry could be affected by multiple physiological and experimental conditions. The absence of some neuropeptides (e.g. corazonin, AST-C, inotocin) in the neuropeptidome analysis may thanks to their low abundance in tested samples, relative short half-life period, and unsuitable chromatographic conditions. Different sample collection methods as well as multiple mass spectrometry methods may be helpful for systematically identification of all neuropeptides. We have added more information in the Discussion to clarify this concern (lines 358-365).

4) Source data supplementary Table 1.This table shows the peptide ions corresponding to the neuropeptide precursor genes.Remarks: Please consult the Locusta neuropeptidomics papers of Clynen and Schoofs and name the peptide precursors according to the originally given neuropeptide names.– Accession number 12049.1 has 2 names: DH/PBAN and tryptopyrokinin. This is, however, the precursor of locustapyrokinin-1, which was identified by neuropeptidome analysis by Clynen and Schoofs and before. It is confusing to rename the genes encoding pyrokinin peptides. AYC12049 definitely is a locustapyrokinin precursor, and should be named according to the original discovery of its encoding neuropeptides, i.e. PK, and not renamed into DH/PBAN. PK-encoding genes have expanded in the evolutionary lineage leading to locusts, when compared to other insects. Naming them as PK1, PK 2, etc. reflects better this expansion (instead of giving them different names).– AYC12051.1 seems to be the precursor of the periviscerokinins and should be named PVK or CAPA.

We accepted the reviewer’s comments. We have carefully compared the peptide sequences identified in our study to that in the previously reported work (Clynen and Schoofs, 2009). We have revised the peptide annotation one by one. And AYC12051.1 has been annotated as PVK, AYC12049 has been annotated as Pyrokinin 1, DH/PBAN has been revised as Pyrokinin 4 (see supplement table 1 for details).

– AKN21242.1 is the precursor for LOM-MIP or locustamyoinhibiting peptide and should not be renamed into AST-B.

We have revised AST-B as MIP as suggested.

– Procolin should be proctolin in the source table of neuropeptides.

We have corrected the spelling mistake in the relative tables and figures.

– SNPF should be sNPF.

We have revised the word as suggested.

5) Line 191 and following: The authors did not experimentally demonstrate that the ACP neuropeptide (or HrTH) acts through the predicted ACP receptor. The authors can show this in vitro by expressing the receptor in a cellular expression system and challenge the receptor with synthetic ACP.In addition, to substantiate the mechanism of action of ACP in vivo, the authors should compare the phenotype of the RNAi-knockdown of the ACP receptor (or Crispr-Cas9 knockout) with the flight phenotypes of WT and FABP knockdowns.

We accepted the reviewer’s concern. To further support the essential role of ACP peptide system in regulating long-term flight, we have provided new experimental data for the molecular and functional characterization of ACPR in the revised manuscript. We have recently successfully constructed the homozygous ACPR mutant locust line using the CRISPR/Cas 9 system. The ACPR mutant (ACPR^13/13^) indeed displayed significantly reduced long-term flight activity, which phenocopied the flight performance caused by *ACP* knockout and *FABP* knockdown. The experimental evidence related to molecular and functional identification of ACPR was presented in the revised Figure 3.

6) line 308: The ACP neuropeptide is abundant in the CC (not in the CA) as shown by Clynen and Schoofs (not in this manuscript). The expression of the gene is confined to cells in the brain as shown in this manuscript. Line 308 is about the neuropeptde. Transcription responses refer to the gene expression, which is brain-specific according to Figure 2. Rephrase.

We understand the reviewer’s concern. In the current study, we performed the neuropeptidome analysis in the CC-CA complex instead of CC alone, and the abundance of ACP peptide is much higher in CC-CA complex than in the brain. So, we wrote the sentence as “ACP peptide is highly abundant in the CC-CA of adult locusts”. However, we described the finding shown by Clynen and Schoofs in the subsequent discussion to clarify the tissue distribution of ACP peptide (lines 382-389). Besides, we have revised the inaccurate description as “the *ACP* precursor gene in the brain displays strong transcription responses to prolonged flight.” (lines 346-347).

7) line 322: "Our results show that the ACP precursor gene is specifically produced in the brain" This phrasing is incorrect: all cells/tissues contain the same genome. No single gene is specifically produced in any tissue". Should be specifically expressed.

Thanks, we have revised the inaccurate description as “the *ACP* precursor gene in the brain displays strong transcription responses to prolonged flight.” (lines 390-391).

8) line 329: it is not quite similar. AKH is synthesized in the CC, more particular by cells in the CC's glandular lobes. JH is synthesized in the CA. ACP is synthesised in the brain and transported via the NCC to the storage lobe of the CC, where it is stored (until release) or where it may influence the release of other substances.

We accepted the reviewer’s comments. We have removed the sentence to avoid misunderstanding.

6) line 317: The ACP neuropeptide or HrTH was isolated from the storage lobes of the corpora cardiaca. These storage lobes contain peptide-storing nerve endings of the NCC coming from the brain.

We have revised the sentence as “the ACP peptide was initially isolated from the storage lobes of the CC of migratory locusts and was named locust hypertrehalosemic hormone (Lom-HrTH)” (lines 382-384).

7) line 332: rephrase: where its receptor is expressed. (A circulating neurohormone will act on all its receptor molecules on the surface of cells (and not only on tissue displaying a high receptor expression), while circulating through the body).

Thanks, we have revised the description as reviewer’s suggestion (line 400).

8) line 365: "Based on transcriptome profile analysis, the flight muscle appears to be the main tissue targeted by ACP, although the ACP receptor also has a relatively high expression level in the fat body".This is speculation. ACP will also target its receptor in the fat body, where it will elicit a distinct functional response.Targets of neuropeptides cannot be (exclusively) identified in the context of altered gene expression upon neuropeptide receptor activation. In fact, most neuropeptides act on GPCRs and the responses are usually very rapid (within seconds), while altered gene expression is a process that takes much more time.Immediately downstream of the neuropeptide-receptor interaction, fast processes are more likely to be the immediate result of this interaction. These fast responses include increased/decreased secretion, change of membrane permeability, opening/closing of ion channels etc. The presence of the ACP receptor in the fat body suggests that – like the flight muscles – also the fat body is a target of ACP, if the predicted ACP receptor is the in vivo cognate receptor of ACP.The manuscript only provides very weak evidence (the presence of a predicted ACP receptor) that ACP directly acts on flight muscles.I suggest to delete lines 365 to 380: many interpretations, wrong phrasing. Does not contribute much to the story.

We agree with the reviewer’s comments. We have removed this part of discussion as the reviewer’s suggestion.

9) line 442: ACP has not been identified in this manuscript. The peptide was identified by Siegert long before and the ACP precursor was already identified in the study of Hou et al., 2015.

We agree with the reviewer’s comments. We have revised the sentence as “we demonstrate that the ACP peptide acts as a novel neuroendocrine regulator controlling lipid transport and utilization associated with long-term flight in locusts” (lines 516-518).

Quantitative neuropeptidome analysis:10) line 476: provide the source of the protein database. Where can it be consulted?

We have provided the source of protein database used. Details were shown as: “The extracted MS/MS spectra were searched against a composite database of *Locust migratoria* (3,286 protein sequences, download from NCBI, 2019) and a protein database (containing 17,307 protein sequences, http://www.locustmine.org:8080/locustmine)(Y*ang et al., 2019*) using in-house PEAKS software”(lines 553-555).

11) why was the enzym trypsin set as a parameter in a peptidomics analysis? This is done for tryptic peptide analysis in a classical proteome analysis, but it is not the optimal way of analysing the neuropeptidome. This parameter does not yield the theoretical spectra of mature neuropeptides. Instead, it yields theoretical spectra of theoretic tryptic peptides, which aid in the identification of the corresponding protein. Spectra of tryptic peptides do not reveal the endogenously occurring neuropeptides, which are endogenously processed by other specific processing enzymes.

We have corrected the wrong description for peptidome analysis. Detailed methods were descripted as: “parent ion mass tolerance is 15 ppm, and fragment ion mass tolerance is 0.05 Da; enzyme specificity, none.

12) line 482: what is meant by this? At least two spectra in one sample?

We used the parameter to improve the reliability of identified peptides according to the neuropeptidome analysis (Han et al., 2015, J Proteome Res)

13) What is the mass tolerance for fragmented peptides? Modifications should also include C-terminal amidation, N-terminal pGlu.

We have revised the methods to make it easy to be understood. It has been shown as: “parent ion mass tolerance is 15 ppm, and fragment ion mass tolerance is 0.05 Da; enzyme specificity, none. The following modifications were applied: C-terminal amidation (A, −0.98) and pyroglutamination from Q (P, −17.03), maximum missed cleavages per peptide: 2, and maximum allowed variable PTM per peptide: 2. A fusion target and decoy approach was used to for the estimation of the false discovery rate (FDR) and controlled at ≤1.0% at the peptide level” (lines 558-567).

14) Figure 3 supplement 2: Explain abbreviations. The expression of ACPR is only tissue-specific for the organs/tissues tested. (The gut, Malpighian tubules, trachea, hypodermis, etc were not analysed). I suggest to rephrase in manuscript text as: Compared to the other organs tested, ACPR is highly expressed in flight muscles and fat body.

We have added the explanation for abbreviations in the figure legend and revised the description for tissue-specific expression analysis of ACPR as “Compared to other organs tested, ACPR was highly expressed in the fat body and flight muscle of adult locusts (Figure 3B)” (lines 200-201).

15) line 567: include reference or source of MEGA software.

We have provided the reference for MEGA software used as the reviewer’s suggestion (line 659).

16) Bogerd et al., (1995) showed that flight activity increased steady-state levels of the AKH I and II mRNAs (approximately 2.0 times each) and the AKH III mRNA (approximately 4.2 times) in the corpora cardiaca. Explain why an altered expression of AKH-1 and AKH-III precursor genes is not seen in the present manuscript.

In the current study, we first screened the top ten adult-abundant neuropeptides (including AKH2 and AKH3) for further expression analysis of their precursor gene in response to flight. The expression level of *AKH2* gene significantly increase after 1 h-sustained flight, the result is consistent with previous findings descripted by Bigerd et al., (1995). However, there is no significant changes in *AKH3* gene expression upon sustained flight, although an increased intendency was observed. The discrepancy between the two studies may be attributed to the different sample collection strategy and detection methods used. In the current study, we collected the CC-CA complex for RNA extraction and measured the gene expression levels by qPCR, whereas only CC tissue was used for RNA isolation and northern blot was used for mRNA level analysis in previous work performed by Bigerd et al. We have added more information in the Discussion to clarify this concern (lines 484-490).

17) Many (supplemental) Figures contain abbreviations that are not explained in the Figure legends.

We have provided the explanation for abbreviation in the corresponding figure legends.

[Editors' note: further revisions were suggested prior to acceptance, as described below.]

The reviewers have discussed their reviews with one another, and the Reviewing Editor has drafted this to help you prepare a revised submission.Essential Revisions:Thank you for clarifying the peptidomic analysis in your revised version.1. However it is essential that your analysis makes clear the source of brain region used for the peptidomic analysis. Therefore "CC-CA" should be replaced by "retrocerebral complex" in the entire manuscript.

Thanks, we accept the reviewer’s suggestions. Now we have replaced “CC-CA” with “retrocerebral complex” in the entire manuscript during revision.

2. Also, rewrite lines 370-375, by stating that in the present manuscript the whole retrocerebral complex was used for peptidomic analysis, in contrast to the study of Clynen who analysed the organs of the retrocerebral complex separately.Delete "a little bit different" (it is not a little bit).

Thanks, we have revised the description as: “In the present study, the whole retrocerebral complex of mature adults was used for peptidomic analysis, in contrast to the study of Clynen and Schoofs, (2009) who analyzed the organs of the retrocerebral complex of immature adults separately” (lines 372-375).

3. An important methodology consideration, that is key to interpreting your results has been raised. Please address this in your revision.– The authors have adapted the methods section on mass spectrometry. It is much clearer now, but one major issue remains. It regards the sentence "Neuropeptide identifications were used only if at least two spectra were identified in one sample".This is a criterion that is used for proteomics analyses; it is not at all suited for neuropeptidomics studies. The reason is that neuropeptides are produced by cleavage from larger preproteins, whereby the mature cleaved peptide is kept and the remainder of the protein is degraded. So, if the preprotein contains only one mature neuropeptide sequence, this neuropeptide cannot be identified by the restricted rule of "two spectra in one sample". The authors refer in their comments to the publication of Han et al. in J. Protein Research. But in the discussion of this article on page 4389, Han et al. clearly state that the 19 neuropeptides that they were unable to identify might be caused by the stricter FDR threshold they used (two spectra in one sample). That is the reason that they missed 19 well known bee neuropeptides in their study.Regarding the present revised manuscript one may wonder in fact how the authors were even able to identify the ACP neuropeptide by using this restrictive threshold. Because the MS-spectra were not provided in the MS (I could not find them), I cannot check what other peptide from the same ACP-precursor they identified. This would be strange, because the remainder of the protein is normally broken down upon cleavage of the ACP peptide out of it.So the question is: did the authors use this restrictive threshold? (which might also explain why they were unable to identify some well known locust peptides residing as single peptides in their respective precursors).If yes, then this reviewer would like to know what other peptide they identified from the ACP precursor, as this would contradict with textbook knowledge about neuropeptide processing. In this aspect, the term "lots of new peptide fragments" is unclear. What is meant by "peptide fragments"? Are these fragments of the same mature peptide or fragments of the remainder of the precursor protein, which should in principle have been degraded.Or did they not use this restrictive threshold?

Thanks for the reviewer’s nice comments. Based on the reviewer’s comments, we realize that our description of neuropeptide identifications is inaccuracy in the sentence “Neuropeptide identifications were used only if at least two spectra were identified in one sample”. In fact, this criterion is used for the identification of peptide fragments. Generally, a single peptide fragment could be detected for multiple times by the MS. A spectrum will be formed each time when the peptide fragment is detected by MS. To avoid misunderstanding, we have revised the description as: “The criterion of peptide fragment identifications was used only if at least two spectra were formed in one sample.” The criterion does not mean that two different peptide fragments derived from the same precursor were detected in one sample. Instead, it means that distinct peptide fragment detected for more than two times were identified in the MS. Despite the differences in modification selection, the sequence characterization of peptide fragments in neuropeptidome study is similar to that of proteomics. The threshold used in the current study (two spectra in one sample) could significantly improve the reliability of peptides identified in the MS. In addition, only several of known neuropeptides, not too many, were not detected under current threshold in our study. The absence of these neuropeptides may be attributed to their low abundance in tested samples, relative short half-life period, and unsuitable chromatographic conditions. We think that the criterion used in the current study should be suitable.

To further clarify the reviewer’s concerns, we have provided the spectra for ACP peptide identified in the peptidomic study (see Author response image 2). In detail, six spectra for ACP peptide were formed in the MS analysis of retrocerebral complex. Based on these spectra, we can clearly confirm the existence of ACP peptide in this sample, and no other peptides were identified for ACP precursor. To avoid misunderstanding, we have provided detailed explanations for the criterion used in the neuropeptidome analysis in the method. The details were shown as: “Generally, a single peptide fragment could be detected for several times by the MS. A spectrum will be formed each time when the peptide fragment is detected by MS. The criterion of peptide fragment identifications was used only if at least two spectra were formed in one sample. Neuropeptides were further validated by comparison with predicted neuropeptide precursors and neuropeptidome analysis in the locust (Clynen and Schoofs, 2009, Hou et al., 2015)” (lines 562-567).

New peptide fragments obtained in the peptidome are mostly newly identified peptides, and degraded products of mature peptides, as well as a few remainders of the precursor proteins (see details in supplementary File 1). All peptide fragments were identified using the same threshold.

**Author response image 2. respfig2:** 

[Editors' note: further revisions were suggested prior to acceptance, as described below.]

The manuscript has been improved but there are some remaining issues that need to be addressed, as outlined below:Reviewer #3:I still have a major concern regarding the neuropeptidome analysis, which I explain in detail below.The authors state in their rebuttal : " Based on the reviewer's comments, we realize that our description of neuropeptide identifications is inaccuracy in the sentence "Neuropeptide identifications were used only if at least two spectra were identified in one sample". To avoid misunderstanding, we have revised the description as: The criterion of peptide fragment identifications was used only if at least two spectra were formed in one sample. Generally, a single peptide fragment could be detected for several times by the MS. A spectrum will be formed each time when the peptide fragment is detected by MS. The criterion of peptide fragment identifications was used only if at least two spectra were formed in one sample"In the text, the authors wrote: "The threshold used in the current study (two spectra in one sample) could significantly improve the reliability of peptides identified in the MS. "Reply:1) This statement is wrong. One MSMS spectrum of good quality is sufficient for the identification of a neuropeptide. An MSMS fragmentation spectrum of good quality is a spectrum where the series of y-ions and b-ions are sufficiently complete to cover the whole amino acid sequence of the peptide (which is the case for the spectra nrs 1 and 4 provided by the authors, but not for spectrum 5).You can never state that at least two MSMS spectra of the same neuropeptide are needed. The argument of the authors "more peptide spectra = more reliability" is also not in line with the way data are collected on the mass spec.

We appreciate the reviewer’s valuable comments. After careful consideration, we agree and accept the reviewer’s suggestions. The two peptide spectra criterion could be used for proteomics analyses, but is not suitable for neuropeptidomic studies. Some neuropeptides with lower abundance may be omitted under the stricter criterion. Given this, we have re-analyzed our peptidomic data with the criterion suggested by the reviewer (using one MS/MS spectrum for neuropeptide identification). Under the new criterion, two additional neuropeptides, Pyrokinin 2 and sulfakinin, were detected in the brain samples, and four additional neuropeptides, Corazonin, natalisin, Ryamide, as well as kinin, were identified in the retrocerebral complex samples. Based on these new findings, we have updated relative figures (Figure 1, Figure 1-S1, and Figure 1-S2) and revised the results description as: “In total, 201 and 362 nonredundant peptides (including both mature neuropeptides and their potential degradation products) derived from 37 neuropeptide precursors were identified in Br and RC, respectively (Supplementary File 1). Tissue-specific analysis showed that neuropeptides from 20 precursors were considerably more abundant in the RC, whereas neuropeptides from 16 precursors were more abundant in the Br. The GPB5-derived peptides showed similar abundance levels in Br and RC (Figure 1—figure supplement 1). The abundant levels of neuropeptides in Br and RC between 5th-instar nymphs and adult locusts were further compared by a label-free quantitative strategy. Compared to 5th-instar nymphs, there were 20 and 18 upregulated neuropeptides in the Br and RC of adult locusts, respectively (Figure 1—figure supplement 2), and 10 neuropeptides displayed significantly higher abundance (Log2FC > 1.5) in either Br or RC of adult locusts (Figure 1A and B)” (lines 115-128). Since the newly identified neuropeptides were not included in the top-ten adult-abundant neuropeptide list, the re-analysis of peptidome data did not affect the screening of flight-related neuropeptides and subsequent results and conclusions in this manuscript.

In data-dependent acquisition, which is how the mass spec collects data, you have to take into account the dynamic exclusion parameter, which ensures that an ion with a particular m/z value will not be selected for fragmentation for a number of seconds to avoid that the instrument detects and measures the same ion over and over again during an elution peak. The normal situation with a dynamic exclusion parameter setting at 15s, is that the mass spectrometer detects a single peptide ion, because peptides typically elute over 30s. The detection of many identical ions of the same peptide is the exception rather than the normal situation and should not be set as a rule of confidence. The dynamic exclusion parameter cannot be found in the methods section.So, in the normal situation, only one precursor ion is selected for MSMS fragmentation. Only when the elution peak of that peptide ion is broad, the peptide ion can be selected multiple times over the period of the elution peak. That is why the dynamic range parameter is important. This parameter avoids that the same ion is selected for fragmentation over and over again at the expense of other – less abundant – peptide ions that elute in the very same region.

Thanks for the reviewer’s comments. Although we did not provide the detailed information in the last version of our manuscript, in fact, we have taken into account the dynamic exclusion parameter as 30 s. In this revised version, we have added the detailed description in the Method and materials of our manuscript as followed: “The eluted neuropeptides were injected into the mass spectrometer via a nano-ESI source (Thermo Fisher Scientific). Ion signals were collected in a data-dependent mode and run with the following settings: full scan resolution at 70,000, automatic gain control (AGC) target 3E6; maximum inject time (MIT) 20 ms; scan range m/z 300–1800; MS/MS scans resolution at 17,500; AGC target 1E5; MIT 60 ms; isolation window 2 m/z; normalized collision energy 27; loop count 10; charge exclusion: unassigned, 1, 8, >8; peptide match: preferred; exclude isotopes: on; dynamic exclusion: 30 s; dynamic exclusion with a repeated count: 1. The MS/MS data were acquired in raw files using Xcalibur software (version 2.2, Thermo Fisher Scientific)” (lines 535-543).

The "more than one spectrum in one sample" argument of the authors is thus a reflection of the elution and abundance of a particular peptide ion. It is not at all a criterion for reliability or confidence regarding the MSMS peptide identification. One spectrum of good quality is as reliable as two or three good spectra of the same ion that just elutes over a longer period of time.

We accept the reviewer’s valuable suggestions. As described above, we have re-analyzed our peptidome data.

The omission of neuropeptides represented by a single MSMS spectrum of good quality is thus a mistake.A second mistake is that all the measured neuropeptide degradation products are counted as bona fide neuropeptides (see below under nr 2).The four ACP spectra provided in the rebuttal are all spectra of the same ACP sequence. The excel file also contains degradation products of this peptide. However, there is no reason to assume that the measurement of multiple spectra in one sample or the measurement of peptide degradation products provide a higher reliability. The measurement of degradation products can be due to a technical issue (sample preparation) or a biological issue (in vivo degradation). It does not tell you anything about the reliability of the measurement of the intact neuropeptide.

We appreciate the reviewer’s suggestions. Under the current condition, we cannot determine whether these shorter peptides are endogenous or degradation products of mature neuropeptides. Therefore, we have revised our description as: “In total, 201 and 362 nonredundant peptides (including both mature neuropeptides and their potential degradation products) derived from 37 neuropeptide precursors were identified in Br and RC, respectively (Supplementary File 1)” (lines 115-122).

As described above, we have re-analyzed the peptidome data using one MS/MS spectrum for neuropeptide identification. By comparison, neuropeptides from 37 precursors (36 precursors in the previous results) were identified after adjusting the peptide identification criterion and only neuropeptide Corazonin was newly identified. In detail, two additional neuropeptides (Pyrokinin2 and sulfakinin) were detected in the brain samples, and four additional neuropeptides (Corazonin, natalisin, Ryamide, and kinin) were detected in the retrocerebral complex samples. Besides, degradation products of several neuropeptides were also detected under new identification criterion. Detailed peptide information was shown in Supplementary file 1. Generally, the alteration of peptide identification criterion does not affect the results of comparative neuropeptidomic analysis (newly identified neuropeptides were not contained in top-ten adult-abundant neuropeptides).

2) Results sectionThe authors state: "In total, 177 and 341 nonredundant neuropeptides derived from 36 precursors were identified in Br and RC, respectively (Supplementary File 1).Reply:Now that the authors have explained how they have interpreted the MSMS data, this sentence is completely wrong and will lead to confusion. The authors have not identified that high number of neuropeptides, because this number reflects multiple MSMS spectra resulting from the same neuropeptide (and its degradation products). A neuropeptide is defined as the amino acid sequence located between the dibasic cleavage sites within the neuropeptide precursor. The shorter versions presented in the excel file are not novel neuropeptides but degradation products. It is unclear what the origin of these degradation products is. They may be artifacts resulting from the sample preparation, or they may represent in vivo degradation fragments resulting from the inactivation of the neuropeptide.The authors should rearrange their excel file by clustering the degradation products of the same neuropeptide within a neuropeptide precursor. Here an example for a pyrokinin (PK1)-precursor derived neuropeptide.GAVPAAQFSPRLamideGAVPAAQFSPRLAVPAAQFSPRLamideAVPAAQFSPRVPAAQFSPRLVPAAQFSPRPAAQFSPRLamideAQFSPRLamide

We appreciate the reviewer’s suggestion. We have revised our description as: “In total, 201 and 362 nonredundant peptides (including both mature neuropeptides and their potential degradation products) derived from 37 neuropeptide precursors were identified in Br and RC, respectively (Supplementary File 1) (lines 115-122). In the discussion, the relative content was revised as: “we obtained a lot of non-abundant neuropeptides as well as their potential degradation products produced by 37 precursors in the main neuroendocrine tissues, including brain and retrocerebral complex. The neuropeptides detected here including most of previously identified peptides (Clynen and Schoofs, 2009)”(lines 349-353). In addition, we have re-arranged the excel file as the reviewer’s suggestion (see Supplementary File 1).

3) The authors also state: " In addition, only several of known neuropeptides, not too many, were not detected under the current threshold in our study. The absence of these neuropeptides may be attributed to their low abundance in tested samples, relative short half-life period, and unsuitable chromatographic conditions. We think that the criterion used in the current study should be suitable."Reply:The identification of the neuropeptides is also dependent on the used database for spectrum scanning, as the authors did not rely on de novo identification. Thus, the identification can only be as good as the database against which the spectra are scanned. If the used locust database does not contain the (protein precursor) sequences, or the derived theoretical spectra, particular experimental spectra will not lead to identification, even if they are abundantly present;The second reason is due (i) to their data acquisition setting (parameter not mentioned in the methods) and to (ii) the "two spectra in one sample" rule that the authors wrongly used, a rule that merely reflects long elution times of very highly abundant peptides (see above). This rule is wrong and excludes less abundant neuropeptides with clear but shorter chromatographic elution peaks.

We appreciate the reviewer’s comments. In fact, it is impossible to identify all neuropeptides in a single peptidome study. Now, we have considered all three aspects concerned by the reviewer during peptidome analysis.

1) Both biochemically identified and bioinformatically predicted neuropeptides as well as downloaded protein database were used for peptidome analysis, which may be helpful for better peptide identification. (2) We have taken into account the dynamic exclusion parameter as 30 s. The detailed description was added in the Method and materials of our revised manuscript (lines 535-543). (3) As described above, we have revised our manuscript by data re-analysis using one MS/MS spectrum for neuropeptide identification. Moreover, we have enriched discussion related to peptide identification as: “However, several neuropeptides (e.g. AST-C, inotocin, and AKH4) identified from previous peptidome study and transcriptome data-based prediction, were not found in the current study. The absence of these neuropeptides in the neuropeptidome analysis may thanks to their low abundance in tested samples, relative short half-life period, unsuitable chromatographic condition or data acquisition setting” (lines 354-359).

4) In the excel files, CC-CA has still not been replaced by retrocerebral complex.

We apologize for the mission. We have revised CC-CA as retrocerebral complex in the excel files (see Figure 2-source data, Figure 3-source data, also supplementary file 1).